# Influenza A(H5N8) vaccine induces humoral and cell-mediated immunity against highly pathogenic avian influenza clade 2.3.4.4b A(H5N1) viruses in at-risk individuals

Oona Liedes [1][✉], Arttu Reinholm[2], Nina Ekström[1], Anu Haveri[1], Anna Solastie[1], Saimi Vara[1], Willemijn F. Rijnink [3], Theo M. Bestebroer[3], Mathilde Richard [3], Rory D. de Vries [3], Pinja Jalkanen[2], Erika Lindh[1], Niina Ikonen[1], Alba Grifoni [4], Alessandro Sette [4,5], Terhi Laaksonen[6], Riikka Holopainen[6], Laura Kakkola[2,7], Maija Lappalainen[8], Ritva K. Syrjänen[1,9], Pekka Kolehmainen[2], Ilkka Julkunen[2,7], Hanna Nohynek[1] & Merit Melin [1]

Finland faced an outbreak of highly pathogenic clade 2.3.4.4b A(H5N1) avian influenza in 2023, which spread from wild birds to fur farms. Vaccinations of at-risk individuals began in June 2024 using the MF59-adjuvanted inactivated A(H5N8) vaccine (Seqirus; A/Astrakhan/3212/2020, clade 2.3.4.4b). Here, in an observational study, we assessed vaccine-induced immune responses in occupational at-risk individuals participating in the phase IV trial, including virus-specific antibody ($n$ = 39 individuals) and T-cell ($n$ = 18 individuals) responses. Vaccination elicited functional antibodies against the vaccine virus and two heterologous clade 2.3.4.4b strains associated with outbreaks on Finnish fur farms and dairy cattle in the United States. Among previously unvaccinated individuals, seroprotection rates against the vaccine virus were 83% (95% CI 70–97%) by microneutralization assay (titre ≥20) and 97% (90–100%) by haemagglutination inhibition assay (titre ≥40). In those previously vaccinated against avian influenza, a single dose induced seroprotection. A(H5N8)-specific memory CD4+ T-cell responses were detectable, with ~5-fold increase in IFNγ secretion after two doses. These results demonstrate that the vaccine probably provides cross-protection against circulating H5 clade 2.3.4.4b viruses. EU Clinical Trial Number 2023-509178-44-00.

Highly pathogenic avian influenza (HPAI) clade 2.3.4.4b A(H5Nx) viruses have been expanding their geographic and host range since 2020. These viruses cause outbreaks in wild birds and poultry worldwide, with spillover to mammals occurring at an alarming frequency. Extensive circulation in multiple species already resulted in the acquisition of several traits associated with increased zoonotic potential[1].

In 2023, Finland experienced a widespread outbreak of clade 2.3.4.4b A(H5N1) that caused mass mortalities among wild birds and spread to 71 fur farms[2,3]. The outbreak, caused by multiple introductions from wild birds, led to the culling of ~500,000 fur animals, mainly foxes, arctic foxes and minks, over 6 months. The causative virus was associated with considerable mortality among wild and captive birds across Europe in the same year[2,4,5]. Epidemiological and genomic investigations identified various transmission modes, including environmental contamination, mammal-to-mammal and possibly mechanical transmission, complicating biosecurity-based control[4].

---

Molecular analyses of the A(H5N1) viruses isolated from fur animals revealed multiple amino acid changes in polymerase basic 2 (PB2) and neuraminidase (NA) proteins associated with adaptation to mammalian hosts[2,4]. Despite extensive occupational exposure, no human infections were detected in Finland.

In March 2024, an outbreak of clade 2.3.4.4b A(H5N1) was reported in dairy cattle in the United States, seeded by a single spillover from wild birds and sustained mainly through mechanical transmission (for example, animal movements, contaminated milking equipment)[6]. After over a year of multistate circulation, the outbreak remains ongoing and has been linked to multiple human cases[7].

In 2024, 84 human cases of A(H5N1) were reported globally, including cases from the USA., Australia, Cambodia, Canada, China and Vietnam[8]. Most were linked to contact with sick or dead animals. While A(H5N1) has historically caused high fatality rates, the 2024 case fatality rate was ~5%, with most cases being mild and identified through surveillance[8]. This suggests that zoonotic transmission may be more common than previously recognized[1,8]. Serological studies also indicate low-level transmission among exposed workers[1]. Although human infections are relatively rare, they carry substantial risks due to viral mutations and reassortment with other influenza A viruses. The widespread circulation and expanding host range of HPAI clade 2.3.4.4b A(H5Nx) viruses increase the likelihood of zoonotic spillovers and emergence of pandemic strains. Infection with A(H5N1) in humans can promote mutations associated with mammalian adaptation, including changes in receptor-binding preferences and polymerase activity, as observed in recent severe cases in Canada and the USA[9,10].

To protect occupational risk groups during the fur farm outbreak in Finland, vaccinations were offered to individuals at risk of exposure to HPAI (for example, fur farm workers, poultry workers, public sector veterinarians, bird ringers and laboratory personnel handling A(H5Nx) HPAI viruses or samples)[11,12]. Finland acquired the MF59-adjuvanted zoonotic influenza vaccine based on A/Astrakhan/3212/2020 (A(H5N8), clade 2.3.4.4b), manufactured by Seqirus, as part of the European Union's joint procurement agreement. This vaccine is expected to provide cross-protection against currently circulating clade 2.3.4.4b viruses[13]. Vaccination efforts in Finland commenced on 13 June 2024, when the vaccine received marketing authorization from the European Medical Agency (EMA)[14].

Finland was the first country that started vaccinating risk groups with this vaccine. There are no previous clinical data regarding immunogenicity of this vaccine or protection against clade 2.3.4.4b influenza viruses in humans. In this study, we investigated vaccine-induced immune responses in at-risk individuals who were offered the vaccine. We measured functional and binding antibodies targeting both the vaccine virus and circulating clade 2.3.4.4b A(H5N1) viruses with microneutralization (MN) and haemagglutinin inhibition (HI) assays, and with fluorescent bead-based multiplex immunoassay (FMIA). CD4+ and CD8+ T-cell responses were characterized with activation-induced marker (AIM) assays and by measuring IFNγ secretion.

## Results

### Study population

We enrolled 52 participants between July and September 2024. Participants were primarily laboratory personnel ($n$ = 31) and bird ringers ($n$ = 12), with fewer poultry workers ($n$ = 5) and veterinarians ($n$ = 4) (Fig. 1a). No fur farm workers participated. Of the 52 participants, 40 provided blood samples before vaccination and after both vaccine doses (Fig. 1b), and peripheral blood mononuclear cells (PBMCs) were obtained from 28 of them. However, in 11 cases, the PBMC yield was insufficient for subsequent analyses. Participants' ages ranged from 27 to 77 years and 73% were female. To account for age-related differences in vaccine responses, analyses were restricted to participants aged ≤65 years. In addition, only participants with samples from all scheduled time points were included in the analysis (Fig. 1). The vaccine

doses were administered at a median interval of 28 days. The baseline blood sample was obtained at a median of 3 days before vaccination and the post-vaccination samples at a median of 20 and 21 days after the first and second vaccine doses, respectively (Fig. 1b). Nine study participants had previously received two to six doses of A(H5N1) vaccines in 2009, 2011–2012 and/or 2018 (Fig. 1, Table 1 and Extended Data Table 1).

### Antibody responses targeting the vaccine antigen

We measured functional antibodies targeting the haemagglutinin (HA) antigen of A/Astrakhan/3212/2020 using MN and HI assays. The seroprotection rates (SPRs) were defined as the proportion of participants with MN titre ≥20 and/or HI titre ≥40. Before vaccination, none of the previously unvaccinated participants had measurable neutralizing antibodies (MN ≥ 10) (Fig. 2a and Table 2). Of 30 participants, 6 had detectable HI antibodies (HI ≥ 10), but none reached the seroprotection level[15] (Fig. 2b and Table 2). In contrast, of the participants who had been previously vaccinated with an H5 vaccine (Table 1 and Extended Data Table 1), 2/9 had detectable neutralizing antibodies, and 7/9 had detectable HI antibodies. Notably, 1/9 had an HI titre of 40. In previously unvaccinated participants, a single vaccine dose induced functional antibodies targeting A/Astrakhan/3212/2020, with geometric mean titres (GMTs) of 15 (MN) or 42 (HI) (Fig. 2a,b and Table 2). The antibody levels increased 2.9-fold (MN) and 6.8-fold (HI) (Extended Data Fig. 1a,b). After one dose, 47% (95% confidence interval (CI) 29–65) of the previously unvaccinated participants reached the seroprotection level based on MN and 73% (58–89) based on HI. Following the second dose, antibody levels increased 3.3-fold (MN) or 2.3-fold (HI) compared to antibody levels after the first dose. Post-second-dose SPRs were 83% (70–97%) by MN and 97% (90–100%) by HI. In participants previously vaccinated with A(H5N1) vaccine, a single dose induced functional antibodies targeting A/Astrakhan/3212/2020, with GMTs of 252 (MN) and 273 (HI) (Fig. 2a,b and Table 2). The antibody levels increased 43-fold (MN) and 16-fold (HI) (Extended Data Fig. 1a,b). Antibody levels were significantly higher than in previously unvaccinated participants for both MN and HI ($p$ < 0.0001). After one dose, 100% (66–100%) of previously vaccinated participants reached seroprotection by both MN and HI, and this rate was maintained after the second dose. Antibody levels did not significantly differ between the first and second dose (MN: $p$ = 0.078, HI: $p$ = 0.47). Titres measured by the two assays correlated strongly ($r$ = 0.89, $p$ < 0.0001) (Fig. 2f).

### Antibody responses targeting A(H5N1) viruses from recent outbreaks

To assess responses to viruses associated with recent mammalian outbreaks, we measured antibody responses against two heterologous A(H5N1) clade 2.3.4.4b HPAI viruses: A/blue fox/UH/004/2023 by MN and A/Texas/37/2024 by HI (Fig. 2c,d, Table 2 and Extended Data Table 2). We also measured HA-specific IgG antibodies against A/Michigan/90/2024 (Extended Data Table 2) using FMIA. Before vaccination, 19/30 of previously unvaccinated participants had detectable antibodies against A/blue fox/UH/004/2023, and 11/30 against A/Texas/37/2024. In previously unvaccinated participants, a single vaccination increased antibody levels 3.4-fold against A/blue fox/UH/004/2023 (MN, GMT 34) and 7.1-fold against A/Texas/37/2024 (HI, GMT 53.6) (Fig. 2c,d and Table 2) and induced H5-specific IgG antibodies against A/Michigan/90/2024 (FMIA, GMC 56 FMIA U ml$^{-1}$) (Fig. 2e). The second dose further increased antibody levels 2.1-fold by MN (GMT 70), 2.1-fold by HI (GMT 113) and 3.1-fold by FMIA (GMC 174 FMIA U ml$^{-1}$) (Extended Data Fig. 1c–e).

Among previously vaccinated participants, 6/9 had measurable MN and 7/9 had measurable HI titres before vaccination. Baseline IgG levels were higher in previously vaccinated (76 FMIA U ml$^{-1}$) than in unvaccinated (18 FMIA U ml$^{-1}$, $p$ = 0.0077). A single vaccine dose increased antibody levels 20-fold by MN (GMT 229), 14-fold by HI (GMT 408) and 20-fold by FMIA (GMC 1,520 FMIA U ml$^{-1}$)

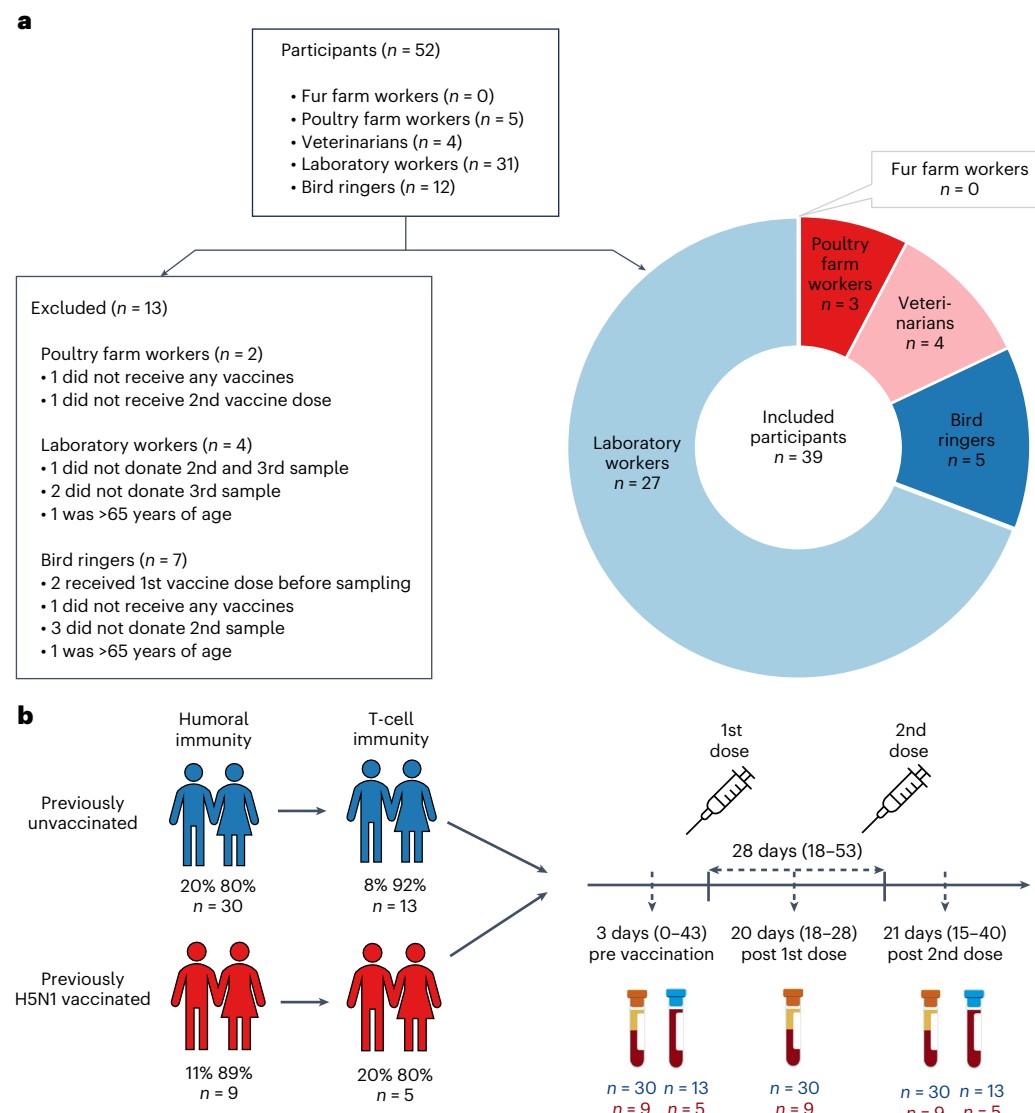

**Fig. 1 | Study overview. a**, Flow diagram illustrating the derivation of the final study cohort by vaccine target group and criteria for exclusion from the analysis. **b**, Timeline of vaccinations and samplings in the study groups of previously unvaccinated (*n* = 30) and A(H5N1) vaccinated (*n* = 9) participants. The distribution of sex (male/female, %) of the participants in each group is indicated. The MF59-adjuvanted A(H5N8) influenza vaccine (clade 2.3.4.4b A/Astrakhan/3212/2020, Seqirus) was administered as a two-dose regimen with a median interval of 28 days (range). Blood was collected for serum separation (red cap) before vaccination, and after the first and second vaccine dose, and for lymphocyte separation (blue cap) before vaccination and after the second vaccine dose (median and range of days and the number of collected serum and cell samples are indicated).

(Extended Data Fig. 1c–e). In previously vaccinated participants, antibody levels did not differ significantly after the first and second doses.

## T-cell responses

To characterize influenza A-specific CD4$^+$ and CD8$^+$ T-cell responses, PBMCs from previously unvaccinated (*n* = 13) and vaccinated (*n* = 5) participants (Extended Data Table 1) were stimulated with overlapping peptide pools covering influenza virus haemagglutinins 1 (H1) and 5 (H5), neuraminidase 8 (N8) and A/Puerto Rico/8/1934 (PR/8) nucleoprotein (NP), and analysed by AIM assay. Representative samples and gating strategy are shown in Figs. 3a and 4a, and Extended Data Fig. 2. Tetanus toxoid and SARS-CoV-2 JN.1 served as positive controls (Extended Data Fig. 3a,b). Before vaccination, 11/13 previously unvaccinated and 5/5 previously vaccinated participants showed CD69$^+$CD134$^+$ CD4$^+$ T-cell responses to H5 and N8 peptides, defined as a ≥2-fold increase over dimethylsulfoxide (DMSO) controls (Fig. 3a,b). In previously unvaccinated participants, CD4$^+$

T-cell responses were observed in 12/13 for H1 and 8/13 for NP, while all previously vaccinated participants responded to both antigens. After two doses, CD4$^+$ T-cell frequencies increased significantly in participants previously unvaccinated for H1 (10/13, 77%, *p* = 0.043) and H5 (11/13, 85%, *p* = 0.0059). In previously vaccinated participants, haemagglutinin-specific CD4$^+$ responses also rose (H1: 4/5, 80%; H5: 5/5, 100%). Fold changes in mean stimulation index (SI) after H5 stimulation were 3.1 in previously unvaccinated and 4.0 in previously vaccinated participants (Extended Data Fig. 4a). Vaccination also increased CD4$^+$ T-cell responses to N8 (9/13, 69%) and NP (9/13, 69%) in previously unvaccinated, and in 2/5 (40%) of previously vaccinated participants for both antigens (Extended Data Fig. 4b).

After two doses, most previously unvaccinated participants showed increased antigen-specific CD69$^+$CD134$^+$ circulating T follicular helper (cTfh) cells (CXCR5$^+$CD45RA$^-$): 7/10 (70%) for H1, 6/9 (67%) for H5 and 6/10 (60%) for N8 (Extended Data Fig. 5a,b). In previously vaccinated participants, antigen-specific cTfh cell responses

**Table 1 | Characteristics of study participants**

| | Samples in analyses of antibody responses | | Samples in analyses of T-cell responses | |
|---|---|---|---|---|
| | **Previously unvaccinated** | **Previously vaccinated** | **Previously unvaccinated** | **Previously vaccinated** |
| *n* | 30 | 9 | 13 | 5 |
| Gender, *n* (%) female | 24 (80%) | 8 (89%) | 12 (92%) | 4 (80%) |
| Age (years), median (range) | 41 (27–63) | 51 (40–61) | 35 (27–54) | 49 (40–51) |
| Target group | | | | |
| Fur farm workers | 0 (0%) | 0 (0%) | 0 (0%) | 0 (0%) |
| Poultry farm workers | 3 (10%) | 0 (0%) | 0 (0%) | 0 (0%) |
| Veterinarians | 4 (13%) | 0 (0%) | 1 (8%) | 0 (0%) |
| Bird ringers | 5 (17%) | 0 (0%) | 0 (0%) | 0 (0%) |
| Laboratory workers | 18 (60%) | 9 (100%) | 12 (92%) | 5 (100%) |
| Dose interval (days), median (range) | 28 (20–53) | 28 (18–42) | 28 (20–53) | 28 (18–35) |
| Number of previous H5 vaccine doses, *n* (%) | | | | |
| 2 | N/A | 4 (45%) | N/A | 2 (40%) |
| 4 | N/A | 2 (22%) | N/A | 1 (20%) |
| 6 | N/A | 3 (33%) | N/A | 2 (40%) |

N/A, not applicable.

increased for H1 in 2/3 (67%), for H5 in 4/4 (100%) and for N8 in 4/5 (80%) participants. Despite these increases, only 1/3, 2/4 and 2/5 participants exceeded the SI cut-off after H1, H5 and N8 stimulation, respectively. The distribution of H5- and N8-specific CD4$^+$ cells across naïve, effector memory (Tem), central memory (Tcm) and effector memory CD45RA$^+$ (TEMRA) subsets remained unchanged from baseline to post-second dose (Extended Data Fig. 5c).

The antigen-specific CD69$^+$/CD137$^+$ CD8$^+$ T-cell responses were assessed by AIM assay using four peptide pools (Extended Data Figs. 4c,d and 6). No significant increase in mean CD8$^+$ responses to any antigen was observed in either group (Extended Data Fig. 6b). Although 8/13 (62%) previously unvaccinated and 3/5 (60%) previously vaccinated participants showed some increase in the frequency of CD69$^+$/CD137$^+$ CD8$^+$ T cells after H5 stimulation, mean fold changes remained low (1.1 and 0.52, respectively) (Extended Data Fig. 4c).

### IFNγ secretion in response to influenza peptides
IFNγ secretion from PBMC supernatants was measured after stimulation with influenza peptide pools (Fig. 4a–d). Stimulation with SARS-CoV-2 JN.1 and tetanus toxoid, used as positive controls, elicited robust responses, confirming that the method reliably detects antigen-specific IFNγ secretion (Fig. 4e,f).

Before vaccination, IFNγ secretion in response to influenza virus antigens was detectable in both unvaccinated and vaccinated participants. Pre-existing T-cell responses were detected in most (11/13) previously unvaccinated and all (5/5) previously vaccinated participants (Fig. 4a). In addition, IFNγ secretion upon N8 stimulation was observed in most previously unvaccinated (12/13) and previously vaccinated (4/5) participants. (Fig. 4c). All participants' cells produced IFNγ upon H1 and NP stimulation (Fig. 4b,d).

After two vaccine doses, IFNγ levels in previously unvaccinated participants increased 5.9-fold and 5.3-fold compared to pre-vaccination values when PBMCs were stimulated with H5 (geometric mean stimulation index (GMSI) 139) or N8 (GMSI 146), respectively (Extended Data Fig. 4e,f). The increase was statistically significant only for N8 ($p = 0.0034$), but not for H5 ($p = 0.24$). Similar increases were observed after stimulation with H1 and NP ($p = 0.15$ and $p = 0.0081$), with significance only for NP.

In previously vaccinated participants, the vaccination increased IFNγ levels by 5.0-fold when stimulated with H5 (GMSI 90) (Extended Data Fig. 4e) and by 3.7-fold when stimulated with

N8 (GMSI 57) (Extended Data Fig. 4f), but these increases were not statistically significant.

We assessed correlations between cellular responses and IFNγ secretion. CD4$^+$ T-cell responses correlated with IFNγ secretion for both H5 ($r = 0.54$, $p = 0.022$; Extended Data Fig. 7a) and N8 peptides ($r = 0.68$, $p = 0.002$; Extended Data Fig. 7c), whereas CD8$^+$ T-cell responses showed no correlation (Extended Data Fig. 7b,d). Antibody titres against A/Astrakhan/3212/202 showed weak, non-significant correlations with IFNγ secretion in response to H5 peptides (MN: $r = 0.30$, $p = 0.23$; HI: $r = 0.27$, $p = 0.27$; Extended Data Fig. 7e,f).

### Discussion
In this study, we evaluated the immunogenicity of the zoonotic influenza vaccine based on clade 2.3.4.4b A(H5N8) virus, A/Astrakhan/3212/2020. Finland was the first country to offer this vaccine to at-risk occupational groups, creating a unique opportunity to assess immunogenicity in the general population. We found that a two-dose regimen induced strong antibody responses against both the vaccine virus and other clade 2.3.4.4b A(H5N1) viruses associated with recent outbreaks on Finnish fur farms and cattle farms in the USA. After two doses, most participants developed seroprotective antibody levels.

While the immunogenicity of this vaccine has not been studied in humans before, similar SPR values have been reported by the manufacturer in clinical trials of earlier vaccine compositions, based on data submitted for marketing authorization[16]. SPR measured by single radial haemolysis ranged from 85% (79–91%) to 91% (87–95%) for A/Vietnam/1194/2004 (clade 1) across two studies and was 91% (85–94%) for A/Turkey/1/2005 (clade 2.2.1) in one study[16]. SPR measured by MN for A/Vietnam/1194/2004 was 67% (60–74%) and 65% (58–72%) in the two studies, and 85% (79–90%) for A/Turkey/1/2005 in the third study[16]. The relatively small sample size in our SPR estimates is reflected by the wider confidence intervals, but as the lower bound still remained above 70% with MN and 90% with HI, the results support the vaccine's immunogenicity in this population.

We observed a strong correlation between the MN and HI titres ($r = 0.89$), indicating that both assays measure functional neutralizing antibodies despite differing sensitivities. The HI assay detects antibodies that block receptor binding, whereas the MN assay measures inhibition of viral entry. Differences in SPR are probably influenced by methodological factors. Neutralizing A(H5N1) antibody responses usually follow HI trends[17–19], although not always[20].

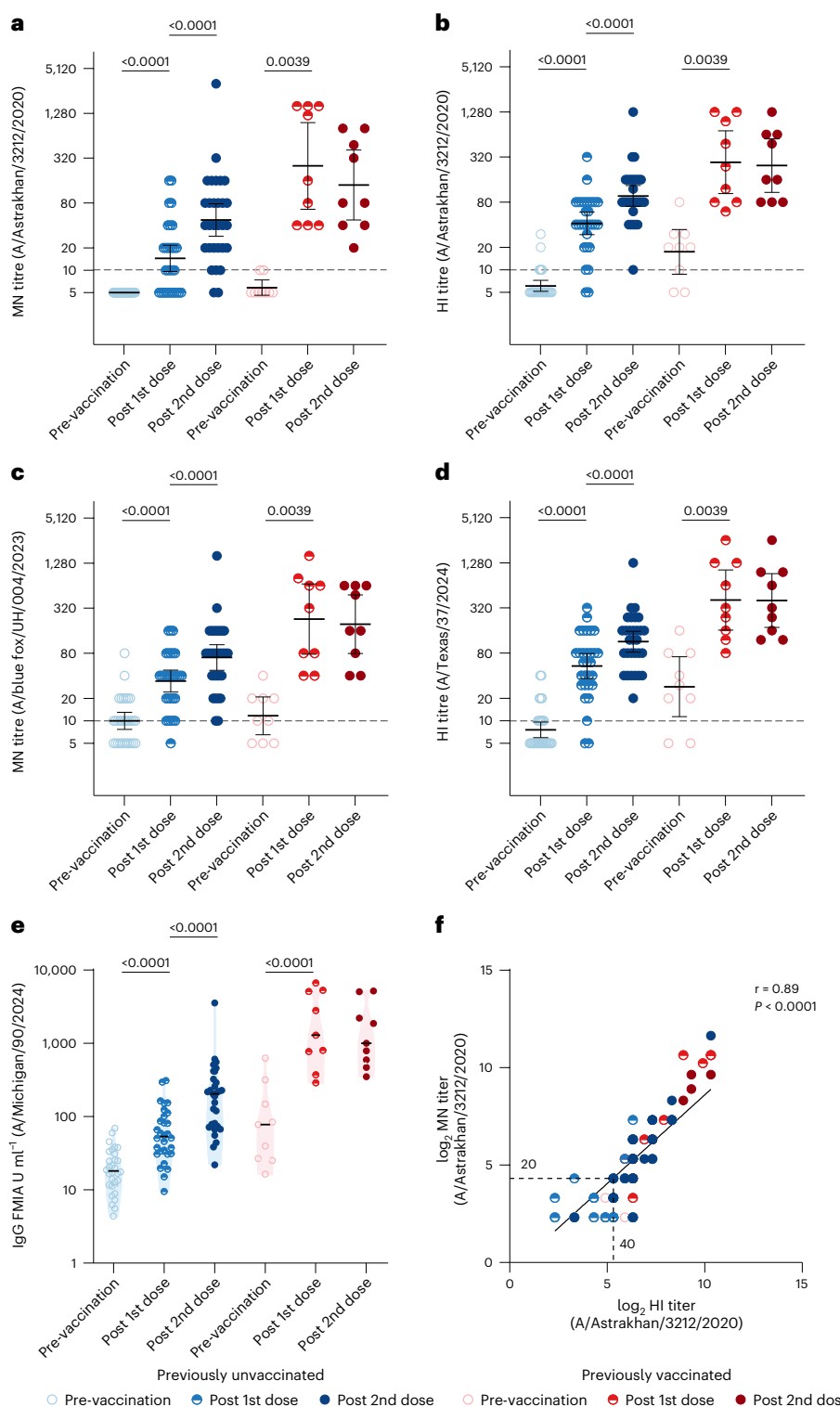

**Fig. 2 | Antibody responses targeting the vaccine antigen and heterologous A(H5N1) clade 2.3.4.4b viruses. a,b**, Antibodies targeting the vaccine antigen A(H5N8) A/Astrakhan/3212/2020 were measured using the microneutralization (MN) assay (**a**) and the haemagglutination inhibition (HI) assay (**b**). **c**, Antibodies targeting A(H5N1) A/blue fox/UH/004/2023 were measured by MN assay. **d**, Antibodies targeting A(H5N1) A/Texas/37/2024 were measured by HI assay. **e**, IgG antibodies binding to purified A(H5N1) A/Michigan/90/2024 haemagglutinin (H5) were measured by fluorescent bead-based multiplex immunoassay (FMIA). Data were categorized into two groups: A(H5N1) unvaccinated ($n = 30$) and previously vaccinated ($n = 9$), at three time points: pre-vaccination, and 3 weeks after the first dose and the second dose. The graphs display geometric mean titres (GMT) and concentrations (GMC) (lines)

and 95% CIs (whiskers). The dashed line indicates the positivity threshold; a titre of 10 or above was considered positive. Exact two-sided $p$ values are reported or shown as $p < 0.0001$ when smaller than the reporting limit of the statistical software. Only statistically significant differences between time points within groups are indicated. Comparisons within a group between two time points were conducted using two-sided Wilcoxon matched-pairs signed-rank test for MN and HI data, and two-sided $t$-test for FMIA data. **f**, Correlation between MN and HI antibody titres to the vaccine antigen A/Astrakhan/3212/2020 was assessed using Spearman's correlation (two-sided, $r = 0.89$, 95% CI 0.84–0.92, $p < 0.0001$) and simple linear regression on $log_2$-transformed data ($R^2 = 0.76$, $p < 0.0001$). Each dot ($n = 142$) may represent values from multiple participants. No error bars are shown.

**Table 2 | Antibody responses to the A(H5N8) vaccine antigen and cross-reactivity of vaccine-induced antibodies with heterologous A(H5N1) clade 2.3.4.4b viruses measured using MN and HI assays and FMIA**

| | | Previously unvaccinated (n=30) | | | Previously vaccinated (n=9) | | |
|---|---|---|---|---|---|---|---|
| | | Pre-vaccination | 3 weeks post 1st dose | 3 weeks post 2nd dose | Pre-vaccination | 3 weeks post 1st dose | 3 weeks post 2nd dose |
| MN (A/Astrakhan/3212/2020) | GMT [95% CI] | 5.0 [5.0–5.0] | 15 [9.6–22] | 47 [29–78] | 5.8 [4.6–7.4] | 252 [66–962] | 140 [47–412] |
| | % positive ≥1:10 (n/n) | 0% (0/30) | 63% (19/30) | 93% (28/30) | 22% (2/9) | 100% (9/9) | 100% (9/9) |
| | % seropositive ≥1:20 (n/n) | 0% (0/30) | 47% (14/30) | 83% (25/30) | 0% (0/9) | 100% (9/9) | 100% (9/9) |
| HI (A/Astrakhan/3212/2020) | GMT [95% CI] | 6.1 [5.2–7.2] | 42 [30–59] | 97 [70–133] | 17 [8.7–35] | 273 [104–718] | 246 [108–560] |
| | % positive ≥1:10 (n/n) | 20% (6/30) | 93% (28/30) | 100% (30/30) | 78% (7/9) | 100% (9/9) | 100% (9/9) |
| | % seropositive ≥1:40 (n/n) | 0% (0/30) | 73% (22/30) | 97% (29/30) | 11% (1/9) | 100% (9/9) | 100% (9/9) |
| MN (A/blue fox/UH/004/2023) | GMT [95% CI] | 10 [7.7–13] | 34 [24–48] | 70 [47–104] | 12 [6.5–21] | 229 [78–676] | 195 [80-480] |
| | % positive ≥1:10 (n/n) | 63% (19/30) | 97% (29/30) | 100% (30/30) | 67% (6/9) | 100% (9/9) | 100% (9/9) |
| | % seropositive ≥1:20 (n/n) | 27% (8/30) | 80% (24/30) | 93% (28/30) | 44% (4/9) | 100% (9/9) | 100% (9/9) |
| HI (A/Texas/37/2024) | GMT [95% CI] | 7.6 [6.0–9.7] | 54 [37–79] | 113 [82–156] | 29 [11–72] | 408 [162–1029] | 401 [175-916] |
| | % positive ≥1:10 (n/n) | 37% (11/30) | 93% (28/30) | 100% (30/30) | 78% (7/9) | 100% (9/9) | 100% (9/9) |
| | % seropositive ≥1:40 (n/n) | 7% (2/30) | 70% (21/30) | 97% (29/30) | 44% (4/9) | 100% (9/9) | 100% (9/9) |
| FMIA (A/Michigan/90/2024 HA) | GMC [95% CI] | 18 [14–24] | 56 [41–77] | 174 [118–257] | 76 [30–195] | 1520 [613–3790] | 1270 [592–2720] |

Higher HI responses have been observed using horse versus turkey or chicken erythrocytes[21,22]. Accordingly, we found that HI titres against the vaccine antigen were higher than MN titres, contradicting earlier findings suggesting that the MN assay is more sensitive[23]. No theoretical protective MN titre has been established, as it varies with virus and method[24–26].

Adjuvanted H5 vaccines can generate cross-reactive antibodies. Two doses of AS03-adjuvanted A/Indonesia/05/2005 (clade 2.1) induced SPRs of 64% (HI) and 77% (MN) against A/Astrakhan/3212/2020, and three doses of MF59-adjuvanted A/Vietnam/1194/2004 (clade 1) resulted in SPRs of 60% (HI) and 95% (MN)[27]. Individuals primed 6 years earlier with MF59-adjuvanted A/duck/Singapore/1997 (clade 0-like) developed higher frequencies of memory B cells and rapidly produced high titres of neutralizing antibodies against diverse A(H5N1) clades after receiving an A/Vietnam/1194/2004 (clade 1) vaccine[28]. These findings suggest that distant priming establishes a pool of memory B cells responsive to mismatched vaccines. In our earlier work, two primary doses generated strain-specific responses, while a later heterologous dose boosted cross-clade antibodies[29]. We found that in previously vaccinated participants, a single dose of the current vaccine elicited a strong antibody response, with no boost from closely spaced second dose, consistent with our previous study[29], reflecting rapid memory B cell activation. In unvaccinated individuals, a second dose continues the primary response, and longer intervals allow memory maturation for a stronger boost.

Since a single vaccination in previously vaccinated participants elicited high levels of neutralizing antibodies, it is reasonable to assume that the response targeted previously encountered epitopes (immunological recall), which are shared or cross-reactive among A(H5) antigens. Immune responses to influenza virus antigens are influenced by pre-existing immunity[30], a phenomenon known as imprinting[31]. In an epidemic situation, it may be beneficial to administer the two vaccine doses close together to achieve protection quickly. However, if the goal is to achieve the best possible cross-protection against viruses different from the vaccine strain, it might be more appropriate to extend the interval between doses by several weeks or months. In recent years, particularly with COVID-19 vaccines, several studies have shown that a longer interval between vaccine doses results in higher antibody responses[32,33]. Another strategy could be to prime the at-risk individuals with a single H5 dose, so only one booster would be needed in a pandemic situation.

In addition to humoral immunity, we also investigated vaccine-induced T-cell responses, as cellular immunity plays a critical role in long-term protection and may contribute to cross-protection against antigenically drifted or heterologous viruses. Although the number of vaccinees analysed for cell-mediated immunity was relatively small, our results indicate that the inactivated avian influenza H5N8 vaccine induces robust virus-specific CD4+ T-cell responses, whereas CD8+ responses in peripheral blood remained weak. Most vaccinees exhibited vaccine-induced activation of H5- and other virus protein (peptide)-specific CD4+ T cells along with IFNγ secretion in response to peptide stimulation. The moderate to strong positive correlations between CD4+ T-cell responses and IFNγ secretion in response to both H5 and N8 peptides suggest that IFNγ production probably originates primarily from CD4+ T cells. Our findings are well in line with previous studies showing that inactivated influenza vaccines (IIV) primarily induce CD4+ T-cell responses[34,35]. Both study participants with and without previous H5 vaccinations exhibited a modest increase in circulating T follicular helper (cTfh) cells, which are linked to the induction of effective humoral immunity[36]. In our analyses, the predominant memory T cell subsets within CD4+ T cells were central and effector memory phenotypes. Central and effector memory cells have previously been identified as the main subsets activated by IIVs[37]. Furthermore, IIV-induced CD4+ T-cell responses[36], particularly cTfh responses[36], have been shown to correlate with antibody titres. Activation of influenza antigen-specific CD8+ T cells was clearly weaker, which is not surprising since IIVs lack synthesis of viral proteins in host cells and thus the activation of CD8+ responses may remain weak[34,38]. The H1-specific CD4+ response to vaccination indicates the activation of a heterosubtypic immune response, which is well documented in animal models of influenza infection[39–41]. However, heterosubtypic responses are thought to be primarily mediated by CD8+ cells[42,43]. The presence of pre-vaccination H5-specific T-cell responses indicate that cross-reactive T cells pre-exist in the general population. A recent study

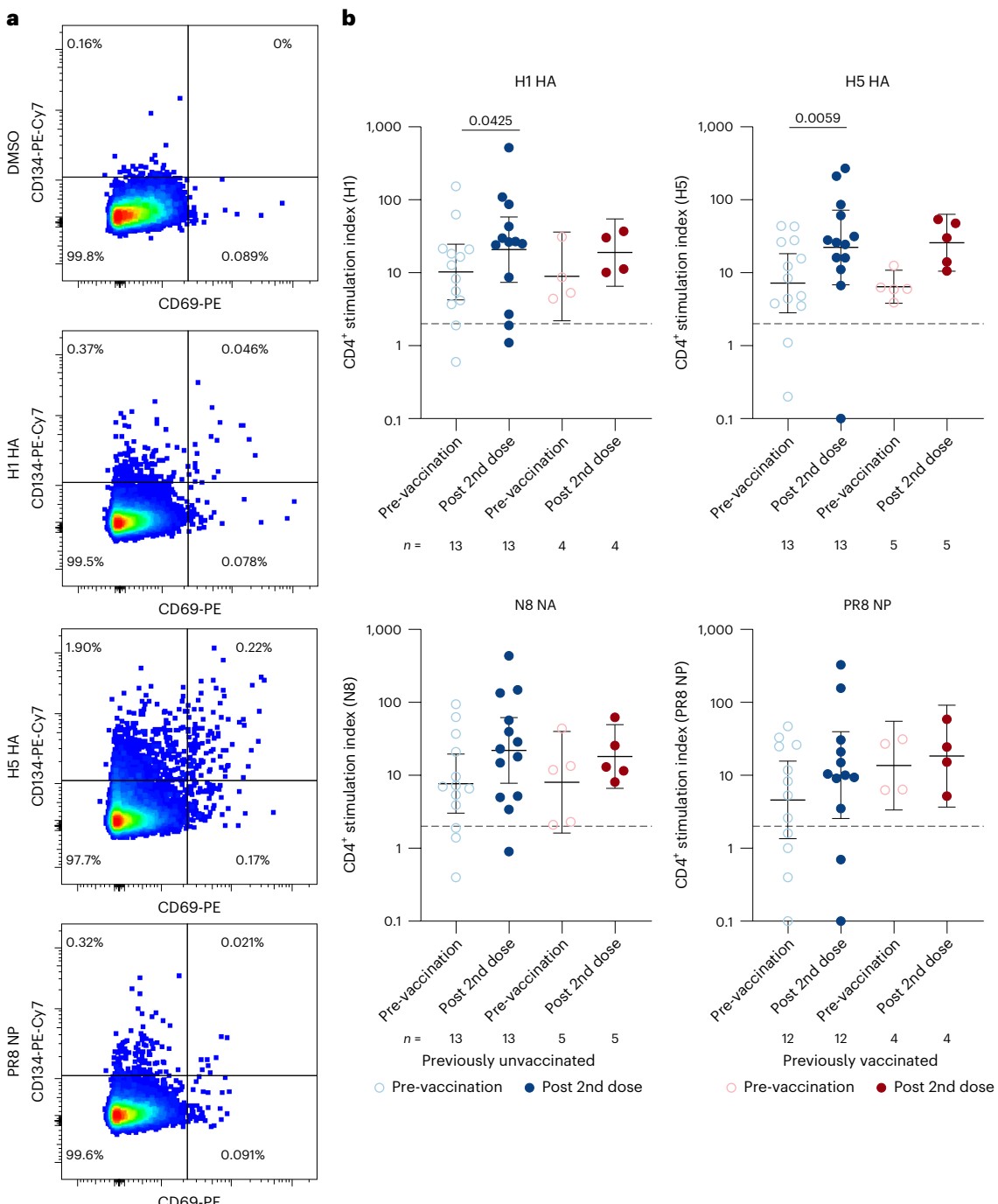

**Fig. 3 | AIM assay for CD4⁺ T-cell responses specific to H1 HA, H5 HA, N8 NA and PR/8 NP after stimulating with corresponding peptide pools. a**, Representative gating for identifying the CD4⁺/CD69⁺/CD134⁺ population. **b**, Fold increases in antigen-specific CD69⁺CD134⁺CD4⁺ T cells in relation to DMSO-stimulated cells. In cases when there were no antigen-specific T cells after DMSO stimulation, the DMSO value of the corresponding pre-vaccination or post-second-dose sample, or the value 0.001 was used. Blue and red dots indicate individuals without and with previous avian influenza vaccinations, respectively. The graphs display geometric mean indices (lines) and 95% CIs (whiskers). Dashed line indicates the cut-off threshold. Statistical significance was determined using two-sided Wilcoxon matched-pairs signed-rank test. Two-sided $p < 0.05$ was considered a significant difference.

on T-cell epitope analysis of A(H5N1) clade 2.3.4.4b suggested that conserved epitopes may enable pre-existing immunity to attenuate the severity of A(H5N1) infections in humans[44]. The authors demonstrated that previous seasonal influenza infections have seeded a broad pool of cross-reactive memory T cells, with ~70% of catalogued CD4⁺ and 60% of CD8⁺ epitopes being ≥90% conserved in circulating clade 2.3.4.4b viruses. Notably, CD4⁺ responses were more pronounced than CD8⁺ responses in a combined AIM and intracellular cytokine staining assay, in line with our findings.

In this study, we measured functional antibodies against the clade 2.3.4.4b A(H5N1) virus detected in dairy cattle in the USA. This specific virus strain was isolated from the first human case during the early phase of the outbreak in March 2024[45,46]. It is important to note that while we observed that the vaccine-induced antibodies effectively recognized this outbreak-related virus strain, it is possible that more recent strains/variants may show impaired neutralization. Mutations in the HA of clade 2.3.4.4b viruses have been reported to occur in the HA head region, which includes the receptor-binding site and surrounding

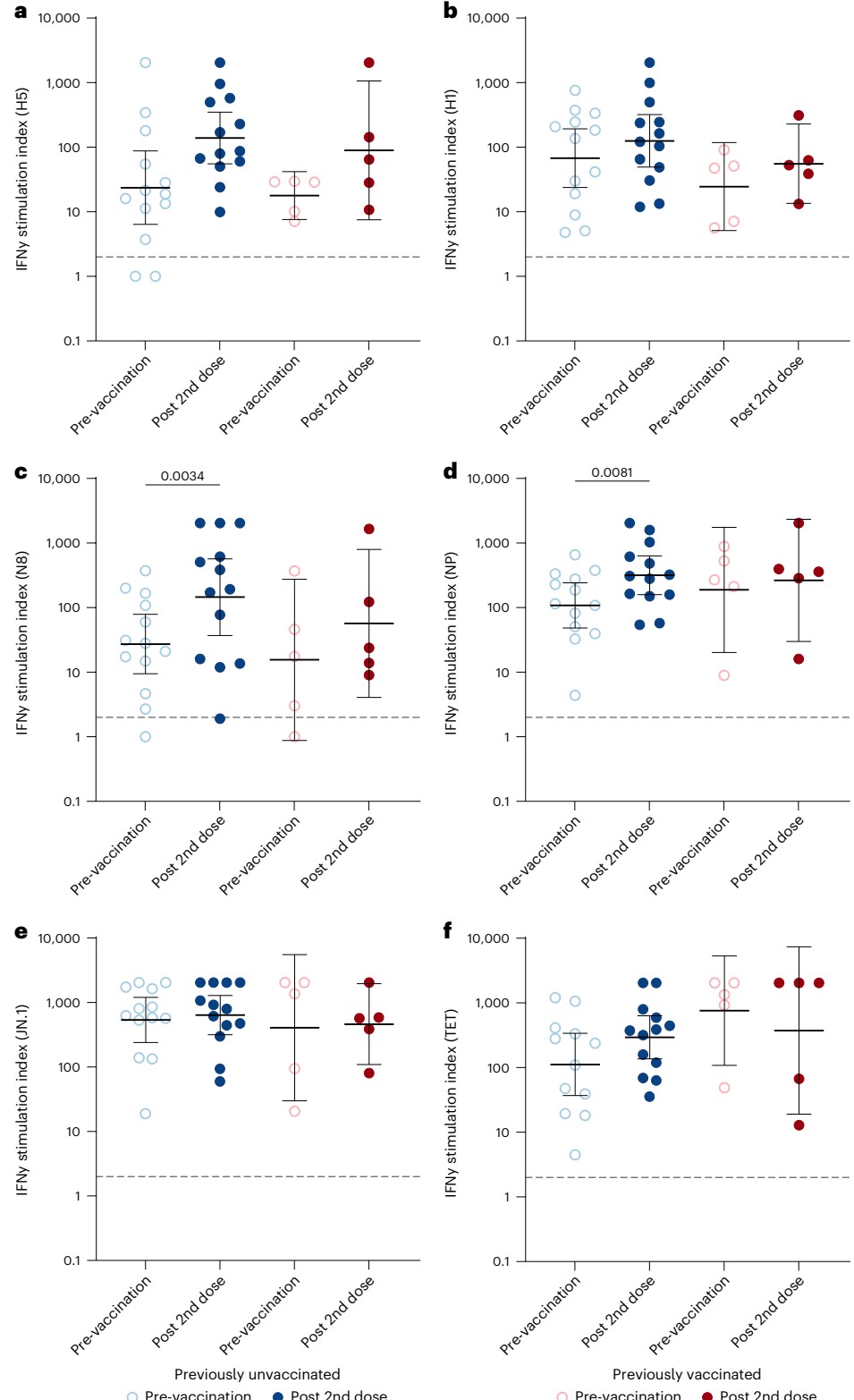

**Fig. 4 | Influenza virus peptide pool-stimulated IFNγ secretion.** Secreted IFNγ from the PBMC supernatants were measured using Luminex assay and the data are presented as stimulation index (SI). **a**, SI values for H5-stimulated PBMC supernatants. **b**, SI values for H1-stimulated PBMC supernatants. **c**, SI values for N8-stimulated PBMC supernatants. **d**, SI values for NP-stimulated PBMC supernatants. As positive controls, PBMCs were stimulated with SARS-CoV-2 JN.1 variant spike peptide pool and tetanus toxoid (TET) **e**, SI values for JN.1 variant spike peptide pool-stimulated PBMC supernatants. **f**, SI values for tetanus toxoid-stimulated PBMC supernatants. IFNγ data were categorized into two groups: previously unvaccinated (*n* = 13) and previously vaccinated (*n* = 5), at two time points: pre-vaccination and 3 weeks after the second dose. The graphs display geometric mean indices (lines) and 95% CIs (whiskers). The dashed line indicates the positivity threshold, which was considered to be SI 2. Only statistically significant differences between time points within groups are indicated. Comparisons within a group between two time points were conducted using two-sided Wilcoxon matched-pairs signed-rank test.

antigenic sites[45,47]. The immunogenicity of the zoonotic influenza vaccine Seqirus A(H5N8) was pre-clinically evaluated in a ferret model[16]. Cross-reactive responses were observed against different clade 2.3.4.4b strains, but no cross-reactivity was detected against A(H5) strains outside clade 2.3.4.4b. In addition, no cross-reactivity was observed for a heterologous strain A/chicken/Ghana/AVL-76321VIR7050-39/2021A(H5N1), despite it being within the same clade 2.3.4.4b as the vaccine.

The final sample size was smaller than planned due to recruitment challenges, largely reflecting the overall low vaccine uptake across all target groups. This was particularly evident among fur farm workers, none of whom participated despite repeated outreach efforts. However, participants were recruited from all other groups covered by the Finnish vaccination recommendation, including laboratory personnel, bird ringers, veterinarians and poultry workers, with laboratory workers forming the largest group. Both participation in the study and receipt of the vaccine were voluntary, which further limited recruitment in groups with low interest in vaccination. Based on our findings, the zoonotic influenza vaccine is expected to confer seroprotection against currently circulating H5 clade 2.3.4.4b viruses. A single dose elicited high neutralizing antibody levels in individuals previously vaccinated against avian influenza, suggesting that priming at-risk individuals with current vaccines may support long-term heterologous immune memory. However, low vaccination coverage among target groups highlights the need for more effective, tailored communication strategies in future vaccination campaigns. Even if the vaccine elicits strong immune responses that are well matched to circulating strains, its overall public health impact ultimately depends on achieving sufficient uptake among those at risk.

## Methods

### Ethical and legal aspects

The study was conducted in accordance with the standards of Good Clinical Practice, the Declaration of Helsinki, and local legal and regulatory requirements, and was registered in the EU Clinical Trial Information System under EU CT number 2023-509178-44-00 on 19 April 2024. The study protocol was authorized by the Finnish Medicines Agency (Fimea) and can be found in the Supplementary Information. Written informed consent to participate was obtained from all participants before sampling. Participation in the study was voluntary and uncompensated.

### Study population and sampling

This observational study was conducted in Finland by the Finnish Institute for Health and Welfare in collaboration with the Finnish Food Authority, HUS Diagnostic Center and University of Turku within the well-being services counties of Helsinki, Uusimaa, Kymenlaakso, southern Carelia, and southern, central and northern Ostrobothnia and Kainuu[48].

We invited individuals to whom the zoonotic influenza vaccine was recommended to participate in the study. Vaccination with the MF59-adjuvanted A(H5N8) influenza vaccine (clade 2.3.4.4b A/Astrakhan/3212/2020, Seqirus)[16] was recommended to those at risk through direct or indirect exposure to infected animals including fur and poultry farm workers, veterinarians, bird ringers and laboratory personnel handling the avian influenza virus or samples that may contain the virus. The national vaccination campaign started in Finland in June 2024. Vaccines were offered in accordance with the national recommendations given by the Finnish Institute for Health and Welfare[49] as a two-dose regimen with a minimum dose interval of 3 weeks. The vaccines were administered through routine healthcare services[12].

The inclusion criteria for the study were: (1) age of 18–65 years, (2) belonging to the target group of the avian influenza vaccine, (3) intention to accept at least one dose of the avian influenza vaccine, (4) a native speaker of Finnish, Swedish or English, (5) home address in

Finland, (6) ability to give samples 3 weeks after each dose, (7) preferably the ability to also participate in the follow-up samplings and (8) a written informed consent. The exclusion criteria were any medical contraindications to influenza vaccination and a history of anaphylactic reaction to any of the constituents or trace residues of the vaccine.

We invited all registered fur and poultry farmers in the well-being services counties of southern, central and northern Ostrobothnia and Kainuu by mail. Farmers were asked to forward invitation letters to their employees. We approached public sector veterinarians, bird ringers and laboratory workers at the Finnish Food Authority, Finnish Institute for Health and Welfare, Helsinki University Hospital and Diagnostic Center, Turku University Hospital and University of Turku, by sending an information letter about the study, and subsequently an invitation letter to those who expressed their interest to participate in the study. Participants were asked to donate a blood sample at their local laboratory center of the well-being services county during three study visits: baseline (within 14 days before the first vaccine dose) and 18–24 days after the first and second vaccine doses. We included two cohorts in the study: (1) participants belonging to the target groups for whom the avian influenza vaccine is recommended (targeted sample size 300) with no previous influenza (A)H5 vaccination history and (2) participants from cohort 1 who have previously received H5 influenza vaccines in 2009, 2011–2012 and/or 2018 (Fig. 1b and Extended Data Table 1).

The targeted sample size of 300 for the study cohort 1 was determined using the sample size formula:

$$n = \frac{Z^2 \times p(1-p)}{E^2} \tag{1}$$

The calculation was based on a desired 95% confidence level ($Z$), an assumed seroprotection rate (SPR) of 75% ($p$) and a 5% margin of error ($E$). The result indicated a minimum sample size of 288 participants required to accurately estimate the proportion of participants achieving seroprotection. With this sample size, the lower limit of the 95% CI is ≥70%. The number of participants recruited to the study in 2024 remained significantly lower, which introduces uncertainty into the seroprotection assessment in this study.

We retrieved contact information of fur and poultry farmers from the Central Database for Animal Keepers and Establishments maintained by the Finnish Food Authority. Information on avian influenza vaccinations given during the study was retrieved from the Register of Primary Health Care Visits. Participants were additionally asked to submit information on previous avian influenza vaccinations, which had been previously recommended for a limited target group of laboratory workers and veterinarians in Finland. The vaccines used in 2009, 2011–2012 and 2018 were the pre-pandemic A(H5N1), inactivated, AS03-adjuvanted A/Indonesia/5/2005 (clade 2.1.3.2)-like split virion vaccine (3.75 µg HA, GlaxoSmithKline); A(H5N1), inactivated, adjuvant-free A/Vietnam/1203/2004 (clade 1)-like whole virus vaccine, (7.5 µg HA, Baxter) and A(H5N1), inactivated, MF59-adjuvanted A/turkey/Turkey/1/2005 (clade 2.2.1)-like strain (NIBRG-23) vaccine (7.5 µg HA, Novartis), respectively.

Serum samples were collected at baseline and after each vaccine dose from all participants. Blood for isolation of PBMCs was additionally collected from laboratory workers and veterinarians in Helsinki and Turku at baseline and after the second vaccine dose. All participants gave written informed consent before the collection of the first study sample.

The study was classified as a low-intervention clinical trial, as the only intervention was the collection of blood samples. No randomization was applied; all samples that met the predefined inclusion criteria were included in the analysis. Investigators were blinded to the identity of participants during all immunological analyses. For FMIA and HI assays, investigators were also blinded to the timing of sample collection with respect to vaccination. For microneutralization and

cellular immunity assays, samples from different time points of the same individual were analysed in parallel within the same run to ensure comparability; therefore, the timing of these samples (pre vs post vaccination) was known to the investigators.

## Cell culture

**Cells for microneutralization (MN) assay.** Madin–Darby canine kidney (MDCK) cells (ATCC-CCL-34, 1805449) were maintained in Eagle's minimal essential medium with L-glutamine (L-Glu) and Earle's balanced salt solution (EMEM, Gibco 6110087), containing 10% fetal bovine serum (FBS, Sigma-Aldrich), 1× non-essential amino acids (NEAA, Sigma-Aldrich), 1.1 g l$^{-1}$ sodium hydrogen carbonate (CHNaCO$_3$, Merck), 100 IU ml$^{-1}$ penicillin (Pen, Sigma-Aldrich) and 100 mg ml$^{-1}$ streptomycin (Strep, Sigma-Aldrich). Cells were tested to be mycoplasma negative, maintained at 37 °C at 5% CO$_2$ and passaged twice per week.

**Cells for HI assay.** MDCK cells (ATCC-CRL-2935) were maintained in EMEM (Capricorn Scientific) with Earle's balanced salt solution, containing 10% FBS, 1× NEAA (Capricorn Scientific), 1.5 mg ml$^{-1}$ sodium bicarbonate (NaHCO$_3$, Gibco), 10 mM 4-(2-hydroxyethyl) piperazine-1-ethane-sulfonic acid (HEPES, Capricorn Scientific), 100 IU ml$^{-1}$ Pen (Capricorn Scientific), 100 mg ml$^{-1}$ Strep (Capricorn Scientific) and 2 mM L-Glu (Capricorn Scientific). Human epithelial 293T cells (ATCC-CRL-3216) were maintained in Dulbecco modified Eagle's medium, high glucose 4.5 g l$^{-1}$ (DMEM, Capricorn Scientific) comprising 10% FBS, 1× NEAA, 1 mM sodium pyruvate (Gibco) supplemented with 2 mM L-Glu, 100 IU ml$^{-1}$ Pen and 100 mg ml$^{-1}$ Strep. Cells were tested to be mycoplasma negative, maintained at 37 °C at 5% CO$_2$ and passaged twice per week (MDCK cells when confluent and 293T cells when sub-confluent). For 293T cells, 500 mg ml$^{-1}$ geneticin (Gibco) was added to the medium during basal cell culture.

## Viruses and antigens

Avian influenza virus strains used in MN and HI assays, and HA antigen used for FMIA are listed with background information in Extended Data Table 2. Antigens used as stimulants in the AIM assays are described below.

## Virus propagation for MN

The A(H5N8) A/Astrakhan/3212/2020 candidate vaccine virus (CVV) with a modified protease cleavage site consistent with a low pathogenic phenotype (IDCDC-RG71A) was received by the Crick Worldwide Influenza Centre, London. The A(H5N1) A/blue fox/UH/004/2023 virus was isolated from a blue fox nasal sample during an outbreak in fur animals in Finland in 2023[50].

Virus strains used in MN assay were further propagated in MDCK cells and collected at the time of cytopathic effect between 50 and 75%. A tissue culture infectious dose of 50% (TCID$_{50}$) was determined and calculated using the Reed–Muench method for each virus stock separately[51] employing the same modified protocol as in the MN assay described below.

## Generation of plasmids and recombinant viruses for the HI assay

**Plasmids.** The A(H5N1) A/Texas/37/2024 virus was isolated from a dairy farm worker in the USA during the cattle outbreak in 2024[52]. The HA segment of A(H5N8) A/Astrakhan/3212/2020 was synthesized by Proteogenix. The HA genes were cloned into a reverse genetics plasmid (modified version of pHW2000) using the GeneArt Seamless Cloning kit (Thermo Fisher)[53].

**Recombinant virus production and sequencing.** Recombinant viruses were produced using the eight-plasmid rescue system[53]. For the HI assay, recombinant viruses carrying seven gene segments of PR/8 high yield (HY)[54] and the A(H5) HA segment of interest, without the

multibasic cleavage site, were generated under biosafety level 2 (BSL2) conditions. Following virus rescue, virus production was evaluated using an HA assay with 1% turkey red blood cells (tRBCs) in phosphate buffered saline (PBS). Virus stocks were propagated in MDCK cells twice and HA gene sequences were verified by Sanger sequencing using the 3500xL Genetic Analyzer (Applied Biosystems). Accession numbers can be found in Extended Data Table 2.

## MN assay

An enzyme-linked immunosorbent assay (ELISA)-based MN assay[26,29,51] was further optimized for the conjugate and substrate steps in this study. Duplicate (technical replicate) heat-inactivated (56 °C, 30 min) serum samples were 2-fold serially diluted starting at 1:10 dilution in MN medium comprising OptiPro SFM medium (Gibco), supplemented with 0.2% bovine serum albumin (BSA), 1× NEAA, Pen and Strep in a total volume of 50 µl. An equal volume of pre-titrated virus was added to obtain 100× TCID$_{50}$ per well, following incubation for 1 h at 37 °C at 5% CO$_2$. MDCK cells were detached, counted and added in a total volume of 100 µl ($2.5 \times 10^4$ cells per well), and the 96-well flat-base tissue culture plates (Sarstedt) were incubated at 37 °C at 5% CO$_2$ for 18–20 h. Wells were washed once with PBS and fixed with ice-cold 80% acetone for 10 min.

The presence of influenza A virus in infected cells was detected by ELISA. Fixed plates were washed twice with washing buffer consisting of PBS containing 0.05% Tween 20 (Sigma-Aldrich). A horseradish peroxidase-labelled (HRP Conjugation Kit - Lightning-Link, Abcam) influenza A nucleoprotein-specific antibody (A7307, Medix Biochemica) was diluted 1:10,000 in PBS containing 5% milk and incubated (80 µl per well) at room temperature for 1 h. After washing six times with the washing buffer, 100 µl of substrate (1-Step TMB ELISA Substrate Solutions, Thermo Scientific) was added into each well and incubated at room temperature for 20 min in the dark. The reaction was stopped with 100 µl 2 N sulfuric acid. Absorbances were measured within 30 min at 450 nm and 620 nm.

The neutralizing endpoint was determined for each individual plate using the following equation[51]:

$$\times = \text{(average OD}_{450} \text{ of virus control wells)}$$
$$\frac{+\text{(average OD}_{450} \text{ of cell control wells)}}{2} \qquad (2)$$

Results were expressed as titres corresponding to the reciprocal of the serum dilution that inhibited 50% of influenza infection. MN titre ≥10 was considered positive, and negative when it was <10. If the titre was <10, a titre of 5 was assigned for statistical calculations.

## HI assay

Recombinant avian influenza viruses in the PR/8 HY background were tested using horse red blood cells (hRBCs) obtained from Cerba Research, Rotterdam, the Netherlands. hRBCs were used instead of the more commonly utilized tRBCs, due to their nearly exclusive expression of α2,3-sialic acid receptors on their surface, which are preferentially bound by avian influenza viruses[21].

After collection, horse blood in citrate buffer was stored at 4 °C for up to 1 month. Before use, hRBCs were washed three times with PBS for 10 min at room temperature, followed by centrifugation at 754 × *g*. Final concentrations of 2% and 10% hRBCs were made in PBS.

Serum samples were absorbed with an equal volume of 10% hRBCs at 4 °C for 1 h, with mixing every 20 min to prevent non-specific agglutination. Subsequently, non-specific inhibition was avoided by incubating sera with in-house manufactured *Vibrio cholerae* filtrate comprising receptor destroying enzyme (RDE) at a 1:6 ratio (v/v) overnight at 37 °C following RDE inactivation at 56 °C for 1 h.

Post RDE inactivation, 2-fold serial dilutions of sera in 0.5% BSA (Sigma-Aldrich) in PBS (0.5% BSA–PBS) were prepared in 96-well V-bottom microtitre plates (Greiner) starting at a 1:20 dilution in a

total volume of 50 µl. Viruses were adjusted to 4 haemagglutinating units (HAU) in 25 µl in PBS and added to each well. Plates were mixed and incubated at 37 °C for 30 min. Following this, 25 µl of 2% hRBCs was transferred to each well, plates were tapped individually, and HI titres were determined after a 1.5-h incubation at 4 °C. In case there was agglutination in the serum control well(s), the HI assay with the corresponding sera was repeated. Six serum samples were absorbed twice instead of once to remove non-specific agglutination. The HI titres were defined as the reciprocal of the last serum dilution in which hRBC agglutination was partially or completely inhibited. The detection limit entailed an HI titre of 10, which was assigned to those serum samples that revealed partial agglutination in the first well. If the titre was <10, a titre of 5 was assigned as the result. Data are presented on the basis of a single experiment.

## Binding antibodies measured with FMIA

The binding of serum IgG to A(H5) was measured with FMIA adapted from an assay used in detection of SARS-CoV-2 antibodies[55]. Purified, commercially available A(H5N1) A/Michigan/90/2024 HA (REC32116, Native Antigen Company) at a concentration of 100 µg ml⁻¹ was conjugated onto MagPlex-C superparamagnetic carboxylated beads (Luminex). Subsequently, 25 µl of beads diluted in PBS (pH 7.2) were added to black 96-well flat-base plates (Costar 3915, Corning) with 25 µl of serum diluted in PBS (pH 7.2 with 1% BSA, 0.8% polyvinylpyrrolidone, 0.5% poly(vinyl alcohol) and 0.1% Tween-20). The plates were incubated for 1 h. This and all subsequent incubations were performed at room temperature in the dark with shaking at 600 r.p.m. After washing with a magnetic plate washer (405TSRS, BioTek), 50 µl of 1:100 diluted IgG detection antibody (R-phycoerythrin-conjugated AffiniPure goat anti-human IgG Fcγ fragment-specific detection antibody, Jackson ImmunoResearch) was added and plates were incubated for 30 min. Following washing, 80 µl of PBS (pH 7.2) was added and plates were incubated for 5 min. Fluorescence was measured with a MAGPIX System (Luminex). Median fluorescence intensity was converted into FMIA U ml⁻¹ by interpolation from 5-parameter logistic curves (xPONENT v.4.2, Luminex) created from a serially diluted (1:400–1:1,638,400) in-house reference pooled from the serum specimens of the present study. All plates were run in duplicates of in-house reference, blank and two control samples. All samples were analysed in 1:400 and 1:1,600 dilutions in duplicate, and results were calculated as the average of four wells. Samples with fluorescence exceeding the reference serum's linear area were reanalysed using further dilutions.

## Isolation and storage of PBMCs

Peripheral whole blood was collected using BD Vacutainer CPT mononuclear cell preparation tubes containing buffered sodium citrate (BD 362761). A total of 48 ml of whole blood was collected from each participant. PBMCs were isolated according to manufacturer instructions and washed two times with Ficoll salt solution. After isolation, PBMCs were counted with a Scepter 3.0 handheld automated cell counter using a 40-µm sensor. The average number of isolated PBMCs was 38 million. Isolated PBMCs were suspended to a concentration of 10⁶ cells per ml in CryoStor CS10 medium (STEMCELL Technologies) and gradually cooled to −80 °C using a Corning CoolCell Freezing Container before being transferred to liquid nitrogen until further use.

## AIM assay and flow cytometry

Avian influenza H1N1 and H5N8 HA, H5N8 NA and PR/8 NP-specific T cells were detected from peptide pool-stimulated PBMCs using an AIM assay as previously described[56]. In addition, the methodology for analysing the proportions of activated antigen-specific follicular T helper cells and memory T cell subtypes (naïve, Tem, Tef, Temra) has also been previously described[57]. PBMCs stored in −150 °C were rapidly thawed in a +37 °C water bath and transferred to 5 ml of culture medium (RPMI-1640, Lonza) supplemented with 10% heat-inactivated human AB serum (Sigma), 2 mM L-Glu (Gibco) and penicillin–streptomycin. PBMCs were washed by centrifuging the cells twice at 600 g for 10 min at +20 °C with fresh culture media. After washing, the viability of the thawed cells was assessed with a TC20 cell counter (Biorad). Cells were plated on a 96-well plate at 10⁶ cells per well in 200 µl of media. Cells were stimulated with DMSO (equimolar, Sigma-Aldrich), tetanus toxoid (20 µg ml⁻¹, AJ vaccines), SARS-CoV-2 JN.1 spike protein (1 µg ml⁻¹, Pepmix, JPT peptides), H1 HA (2 µg ml⁻¹), H5 HA (2 µg ml⁻¹), N8 NA (2 µg ml⁻¹) or PR/8 NP (2 µg ml⁻¹; Pepmix, JPT peptides), peptide pools covering the whole proteins, after which the cells were incubated at 5% CO₂, +37 °C for 72 h. Before choosing the optimal incubation, time and peptide concentrations pretesting was done with H5N8-vaccinated individuals. An equimolar amount of DMSO was used as a negative control stimulus. SARS-CoV-2 JN.1 and PR/8 NP peptide pools were 15-mers with 11-mer overlaps, and H1, H5 and N8 peptide pools were 15-mers with 10-mer overlaps synthetized by TC peptide Lab as crude material. The peptides were then pooled and sequentially lyophilised with the MegaPool approach[58] and resuspended at a stock concentration of 1 mg ml⁻¹.

After stimulation, cells were centrifuged at 600 g for 10 min at +20 °C and washed with FACS 1 buffer (PBS; 0.01% NaN₃), followed by staining the cells with a cell viability dye (Zombie Green, BioLegend) diluted with FACS 1 buffer. After incubating the cells for 15 min in the dark at room temperature, fluorochrome-conjugated antibodies diluted in FACS 2 buffer (PBS, 2% FCS; 0.01% NaN₃) were added (Extended Data Table 3). After incubating the cells for 30 min at +4 °C, cells were washed for 10 min at +20 °C with FACS 2 and FACS 1 buffers, followed by centrifugation at 600 g after each wash. After washing, the cells were fixed for 20 min with 4% formaldehyde in PBS. Finally, cells were washed and suspended with FACS 1 buffer, followed by acquisition with a BD LSRFortessa flow cytometer (BD Biosciences), and results were analysed with FlowJo 10.10.0. The AIM test stimulation index (SI) was calculated by dividing the percentage of peptide pool-stimulated CD4⁺/CD69⁺/CD134⁺ or CD8⁺/CD69⁺/CD137⁺ T cells by the DMSO-stimulated CD4⁺/CD69⁺/CD134⁺ or CD8⁺/CD69⁺/CD137⁺ T cells. Samples with less than 10,000 CD3⁺ cells were excluded from all analyses, and samples with less than 500 circulating T follicular helper (cTfh) CD4⁺ cells were excluded from cTfh cell analysis.

The assay was optimized by incubating PBMCs acquired from two vaccinated individuals pre and post vaccination for 48 and 72 h after stimulating with DMSO (equimolar), TET (20 µg ml⁻¹), SARS-CoV-2 XBB.1.5 spike PepMix™ (0.5 µg ml⁻¹; 1 µg ml⁻¹; 2 µg ml⁻¹; JPT peptides), H5 (1 µg ml⁻¹; 2 µg ml⁻¹, 4 µg ml⁻¹, 8 µg ml⁻¹) and H1 (1 µg ml⁻¹; 2 µg ml⁻¹, 4 µg ml⁻¹, 8 µg ml⁻¹), as described above.

## IFNγ detection in PBMC supernatants using Luminex

The concentration of IFNγ secreted into the supernatants during the 72-h stimulation of the cell cultures was measured using a 96-well plate assay with the MILLIPLEX MAP Kit HCD8MAG-15K (Millipore)[56]. The fluorescence of the samples was measured using the Luminex MAGPIX magnetic bead analyser (Luminex). Samples that were within the linear range of the kit's standard curve were given their measured concentration. Samples below the lowest standard in the linear range were given half the value of the standard (2.4 pg ml⁻¹ for IFNγ), and samples above the highest standard were given the highest value of the standard (5,000 pg ml⁻¹ for IFNγ). Standards with a standard deviation of less than 20% for the duplicates were accepted. According to the kit manufacturer, if there were less than 35 beads in the well, the samples could not be assigned a reliable concentration, hence those samples were discarded from the final analysis. Results were expressed as SI, defined as the ratio of IFNγ concentration after peptide pool or tetanus toxoid stimulation to the corresponding concentration after DMSO stimulation within the same sample. A stimulation response was considered positive if the IFNγ SI value measured in the sample exceeded the positive threshold (2 SI).

## Statistical methods

Data analyses were performed with MS Excel v.2408, GraphPad Prism v.10.2.3 and 10.4.1, R v.4.2.1 and FlowJo v.10.10.0. $P < 0.05$ was considered statistically significant. All results are presented as descriptive statistics; statistical tests were performed without adjustment for multiple comparisons. For intra- and intergroup comparison, data were categorized into two groups according to vaccination history. Only data from participants who provided samples at the three different time points and received vaccinations in the correct order relative to sampling were included in the analysis of vaccine responses. Geometric means and 95% CIs were calculated for the neutralizing antibody titres, IgG antibody concentrations, T cell and IFNγ SIs. Fold changes were calculated from the geometric means of each group.

The Shapiro–Wilk normality test was used to determine whether data were normally distributed. Since MN and HI data were not normally distributed, the data were $\log_2$ transformed and non-parametric tests were used. As T cell data were not normally distributed, non-parametric tests were used. The Mann–Whitney $U$-test was used to compare differences between different groups, while the Wilcoxon matched-pairs signed-rank test was used for within-group comparisons across different time points. As IgG data were normally distributed, comparisons between groups were performed with a $t$-test. Correlation analyses were performed using the non-parametric Spearman rank correlation coefficient.

To assess the correlation of titres against A/Astrakhan/3212/2020 measured with MN and HI tests, data from all 142 samples that had results measured with both tests, including serum samples from participants who did not provide all three samples, were included. An HI titre of 40 is typically accepted to correspond to a 50% or more reduction in the risk of contracting an influenza infection or influenza disease[59] and defined by both the US Food and Drug Administration and the European Medicines Agency Committee for Medicinal Products for Human Use as the primary correlate of protection[60]. To determine the MN titre corresponding to an HI titre of 40 against A/Astrakhan/3212/2020, the data were $\log_2$ transformed, and Spearman's correlation was performed ($r = 0.89$, $p < 0.0001$). The Spearman correlation coefficient indicated a positive correlation between MN and HI titres. To further explore this relationship, regression analysis was conducted. A simple linear regression model was applied to assess the equivalence between MN and HI titres ($R^2 = 0.76$, $p < 0.0001$), yielding the equation $Y = 0.9098x - 0.4967$. Based on this model, an HI titre of 40 corresponds to an MN titre of 20. The percent SPR for each group was calculated as the number of seropositive samples (MN titres ≥20 or HI titres ≥40) divided by the number of samples ×100 in the group. Confidence intervals for SPRs were calculated with normal approximation to the binomial calculation.

## Reporting summary

Further information on research design is available in the Nature Portfolio Reporting Summary linked to this article.

## Data availability

At the outset of the trial, data-sharing provisions were not included in the informed consent documents signed by participants. In accordance with ethics and institutional policies, we are not authorized to release individual-level or pseudo-anonymized datasets to the public. To protect participant privacy, only de-identified, aggregated group-level values (without background or individual-level information) are available. These data can be requested from the corresponding author (oona.liedes@thl.fi) and will typically be provided within 2–4 weeks, subject to review for compliance with applicable ethics requirements. The study protocol is provided as a Supplementary Information file. Source data are provided with this paper.

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

## Acknowledgements

We thank S. Suopanki, L. Hagberg, M. Heikkilä, R. Hanninen, M. Suorsa, A.-M. Pieniniemi and A. Suominen for support with clinical

documentation and sample handling; M. Heikkilä, S. Suopanki, M. Määttä, M.-L. Ollonen, K. Mäkisalo, M. Eskelinen, V. Avelin, E. Altan and P. Österlund for contributions to laboratory work. The nasal sample from a blue fox was kindly provided by T. Sironen (University of Helsinki), and plasmids were generously provided by D. R. Perez (University of Georgia, USA). We acknowledge M. Aura, E. Ruokokoski, J. Oksanen and T. Rauhala for data management, and J. Niemi for sample management; C. Savolainen-Kopra, A. Katz and O. Helve (THL), as well as M. Koopmans (Erasmus MC), for providing facilities and resources that enabled this research; A. A. Palmu, E. Isosaari and H. Nieminen (Finnish Vaccine Research) for guidance on clinical study conduct, coordination and regulatory processes. The use and expertise of the Turku Bioscience Protein Structure and Chemistry Core Facility, a member of Biocenter Finland, FINStruct, and Turku Protein Core, are gratefully acknowledged. We also acknowledge the authors and their laboratories who generated and submitted sequences to GISAID's EpiFlu Database. Finally, we thank all participating laboratory centres and volunteers for essential contributions to this study. The Finnish Institute for Health and Welfare (THL) funded the clinical vaccine study. For the immunological studies, THL and the University of Turku received funding from the Academy of Finland (Avian and seasonal influenza vaccine-induced humoral and cell-mediated immune responses, Decision numbers 362192 and 362193). The study was also funded by the Netherlands Organization for Health Research and Development (ZonMw, NCOH Pandemic Preparedness Research Kickstarter, grant agreement 10710022210003) (R.D.d.V.), European Union's EU4Health programme DURABLE (grant number 101102733) (W.F.R., M.R., R.D.d.V.), and the Dutch Ministries of Agriculture, Fisheries, Food Security and Nature and Health, Welfare and Sport (M.R., R.D.d.V.). The funders had no role in study design, data collection and analysis, decision to publish or preparation of the manuscript.

## Author contributions

M.M., H.N., R.K.S., N.E. and I.J. designed the study. O.L., A.R., A.H., A. Solastie, S.V., P.J., P.K., W.F.R., T.M.B., A.G. and A. Sette conducted the immunological research. O.L., A.R., A.H., A. Solastie, S.V., W.F.R., M.M., M.R. and R.D.d.V. contributed to the data analysis. N.E. and R.K.S. managed the coordination and regulatory process of the clinical study. N.E., T.L., R.H., L.K. and M.L. contributed to the recruitment. O.L., A.H., A. Solastie and S.V. were involved in handling the clinical documents and samples. M.M., O.L., A.R., N.E., A.H., A. Solastie, S.V., W.F.R., R.D.d.V., E.L., P.K., L.K., I.J. and H.N. wrote the manuscript. T.L., R.H., M.L., I.J., L.K., R.K.S., N.I., M.R., A.G. and A. Sette assisted with the editing of the text.

## Funding

## Competing interests

A. Sette is a consultant for Alcimed, Arcturus, Darwin Health, Desna Therapeutics, EmerVax, Gilead Sciences, Guggenheim Securities, Link University and RiverVest Venture Partners. The La Jolla Institute for Immunology (LJI) has filed for patent protection for various aspects of T cell epitope and vaccine design work. R.K.S. has acted or acts as a sub-investigator in a COVID-19 study sponsored by Pfizer, a pneumococcal carriage study sponsored by Merck Sharp & Dohme, and influenza, pertussis and meningitis studies sponsored by Sanofi Pasteur, not related to this work. Her current affiliation, Finnish Vaccine Research (FVR), conducts clinical trials and studies sponsored by almost all vaccine providers, not related to this work. H.N. is a member of the National Immunization Technical Advisory Group, THL, Finland, and the chair of the WHO Strategic Advisory Group of Experts. M.M. is a member of the National Immunization Technical Advisory Group, THL, Finland. O.L., A.R., N.E., A.H., A. Solastie, S.V., W.F.R., T.M.B., M.R., R.D.d.V., P.J., E.L., N.I., A.G., T.L., R.H., L.K., M.L., P.K. and I.J. declare no competing interests.

## Additional information

**Extended data** is available for this paper at https://doi.org/10.1038/s41564-025-02183-5.

**Correspondence and requests for materials** should be addressed to Oona Liedes.

¹Finnish Institute for Health and Welfare, Helsinki, Finland. ²Institute of Biomedicine, Faculty of Medicine, University of Turku, Turku, Finland. ³Department of Viroscience, Erasmus University Medical Center, Rotterdam, the Netherlands. ⁴Center for Vaccine Innovation, La Jolla Institute for Immunology (LJI), La Jolla, CA, USA. ⁵Department of Pathology, University of California, San Diego, La Jolla, CA, USA. ⁶Finnish Food Authority, Helsinki, Finland. ⁷Clinical Microbiology, Turku University Hospital, Turku, Finland. ⁸HUS Diagnostic Center, Helsinki University Hospital, Helsinki, Finland. ⁹FVR – Finnish Vaccine Research, Tampere, Finland. ✉e-mail: oona.liedes@thl.fi

**Extended Data Table 1 | Avian influenza vaccines administered in previously vaccinated participants**

| | Pre-pandemic A(H5N1), inactivated, AS03-adjuvanted A/Indonesia/5/2005-like split virion vaccine, clade 2.1.3.2 (GlaxoSmithKline) | | A(H5N1) inactivated adjuvant-free A/Vietnam/1203/2004-like whole virus vaccine, clade 1 (Baxter) | | A(H5N1) inactivated MF59-adjuvanted A/turkey/Turkey/1/2005 (H5N1)-like strain (NIBRG-23) vaccine, clade 2.2.1 (Novartis) | | Zoonotic A(H5N8), inactivated, M59-adjuvanted, A/Astrakhan/3212/2020-like strain (CBER-RG8A) vaccine, (clade 2.3.4.4b) (Seqirus Vaccines) | |
|---|---|---|---|---|---|---|---|---|
| | 2009 | | 2011-2012 | | 2018 | | 2024 | |
| Participant | 1st dose | 2nd dose | 1st dose | 2nd dose | 1st dose | 2nd dose | 1st dose | 2nd dose |
| 1 | | | | | • | • | • | • |
| 2 | | | | | • | • | • | • |
| 3 | • | • | • | • | | | • | • |
| 4 | • | • | • | • | • | • | • | • |
| 5 | • | • | • | • | • | • | • | • |
| 6 | • | • | • | • | • | • | • | • |
| 7 | | | | | • | • | • | • |
| 8 | • | • | | | • | • | • | • |
| 9 | | | | | • | • | • | • |

Detailed information and the number of A(H5) vaccine doses given prior to and during this study indicated by year for each participant. Vaccination is indicated by a solid circle (•).

**Extended Data Table 2 | Viruses and antigens used in the microneutralization (MN) assay, the hemagglutination inhibition (HI) assay, and the fluorescent bead-based multiplex immunoassay (FMIA)**

| Virus or antigen name | GISAID isolate ID | HA accession number | Subtype | HA clade | Full virus or RG or HA | Passage history | Origin of the virus or HA | Biosafety level handling | Method |
|---|---|---|---|---|---|---|---|---|---|
| A/Astrakhan/3212/2020 CVV | EPI_ISL_13655139 | EPI2084527 | A(H5N8) | 2.3.4.4.b | IDCDC-RG71A | E1M2 | The Crick Worldwide Influenza Centre, London (Ex/E1) | BSL2+ | MN |
| A/Astrakhan/3212/2020 | EPI_ISL_1038924 | EPI1846961 | A(H5N8) | 2.3.4.4.b | 7+1 PR/8 HY | 293TM2 | Gene synthesis from Proteogenix | BSL2 | HI |
| A/blue fox/UH/004/2023 | EPI_ISL_18764855 (E1 passage) | EPI2914899 (E1 passage) | A(H5N1) | 2.3.4.4.b | Full virus | M2 | Associate Professor Tarja Sironen, University of Helsinki | BSL3 | MN |
| A/Texas/37/2024 | EPI_ISL_19027114 | EPI3171488 | A(H5N1) | 2.3.4.4.b | 7+1 PR/8 HY | 293TM2 | Plasmid from Dr. Daniel R. Perez, University of Georgia, US | BSL2 | HI |
| A/Michigan/90/2024 | EPI_ISL_19162802 | EPI3334182 | A(H5N1) | 2.3.4.4.b | HA amino acids 1-530 | Original | Antigen from Native Antigen Company | BSL1 | FMIA |

CVV = candidate vaccine virus, HA = hemagglutinin, RG = reverse genetics, PR/8 HY = A/Puerto Rico/8/1934 high yield, M = MDCK.

**Extended Data Table 3 | Fluorochorome-antibody panel used in the flow cytometry**

| Antibody | Fluorochrome | Amount used | Manufacturer | Cat# |
|---|---|---|---|---|
| Anti-human CD45 (HI30 clone) | APC-eFluor780 | 2 µl/test | Invitrogen/Life technologies | 47-0459-42 |
| Anti-human CD3 (UCHT1 clone) | eFluor506 | 5 µl/test | Invitrogen/Life technologies | 69-0038-42 |
| Anti-human CD4 (RPA-T4 clone) | eFluor450 | 5 µl/test | Invitrogen/Life technologies | 48-0049-42 |
| Anti-human CD8a (SK1) | PerCP-eFluor710 | 5 µl/test | Invitrogen/Life technologies | 46-0087-42 |
| Anti-human CD69 (FN50 clone) | PE | 15 µl/test | BD Biosciences | 555531 |
| Anti-human CD134 (ACT35 clone) | PE/Cyanine7 | 5 µl/test | BioLegend | 350012 |
| Anti-human CD137 (4B4-1 clone) | APC | 5 µl/test | BioLegend | 309810 |
| Anti-human CD45RA (HI100 clone) | Brilliant Violet 785 | 2 µl/test | BioLegend | 304140 |
| Anti-human CD197 (CCR7) (G043H7 clone) | PE/Dazzle 594 | 5 µl/test | BioLegend | 353236 |
| Anti-human CD185 (CXCR5) (J252D4 clone) | Brilliant Violet 605 | 5 µl/test | BioLegend | 356930 |

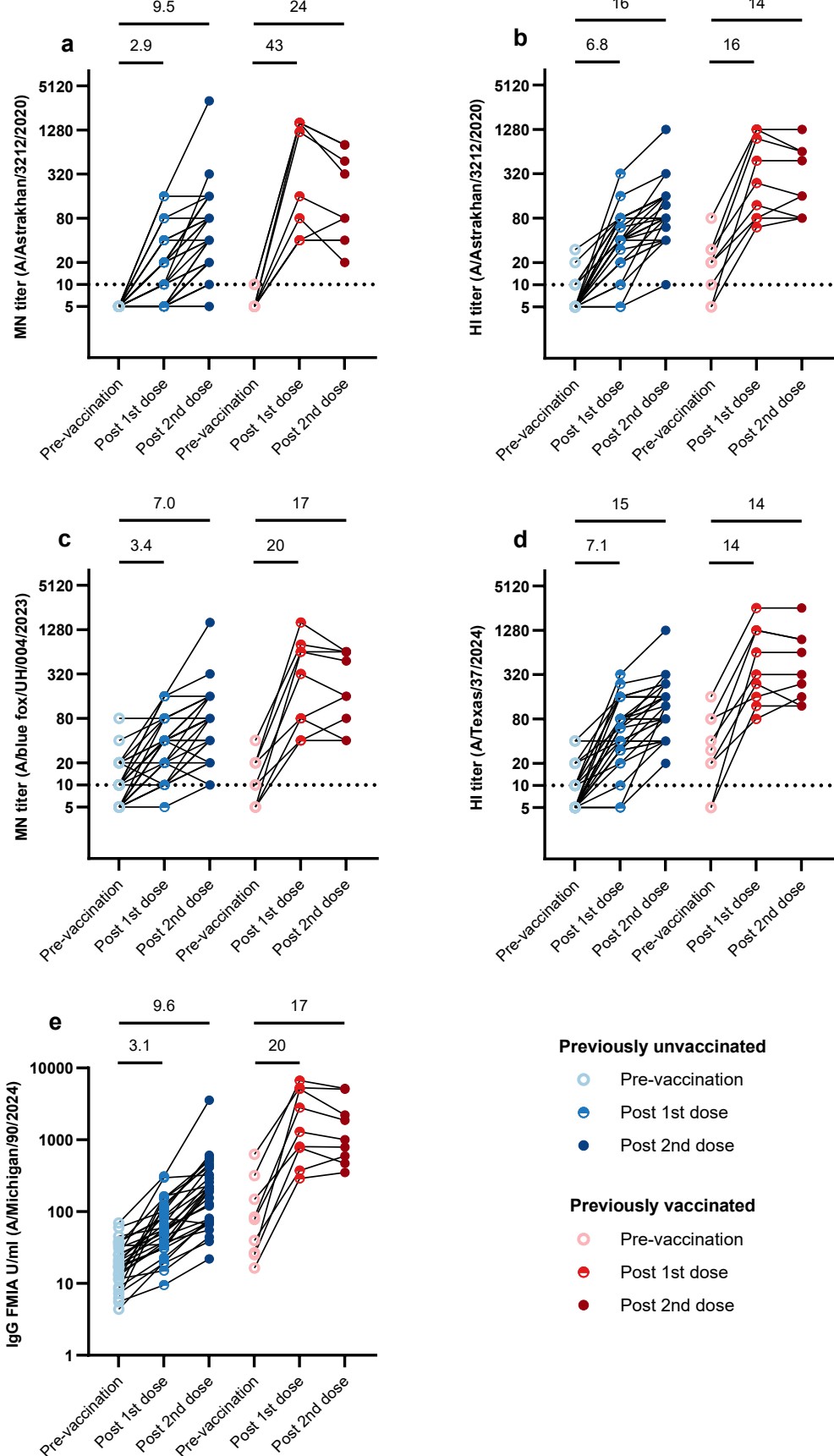

**Extended Data Fig. 1 | See next page for caption.**

**Extended Data Fig. 1 | Kinetics of vaccine-induced individual antibody responses. a**, **b**, Antibodies targeting the vaccine antigen A(H5N8) A/Astrakhan/3212/2020 were measured using the microneutralization (MN) and the hemagglutination inhibition (HI) assay. **c**, Antibodies targeting A(H5N1) A/blue fox/UH/004/2023 were measured by the MN assay. **d**, Antibodies targeting A(H5N1) A/Texas/37/2024 were measured by HI assay. **e**, IgG antibodies binding to purified A(H5N1) A/Michigan/90/2024 H5 type HA antigen were measured by fluorescent bead-based multiplex immunoassay (FMIA). Individual responses are shown as lines for two groups, A(H5N1) unvaccinated (n = 30) and previously vaccinated (n = 9), at three time points: pre-vaccination, and three weeks after the first dose and the second dose. The dashed line indicates the positivity threshold. Fold changes in the mean antibody titers before vaccination and after the first dose, as well as before vaccination and after the second dose, represented by lines within the graphs are shown.

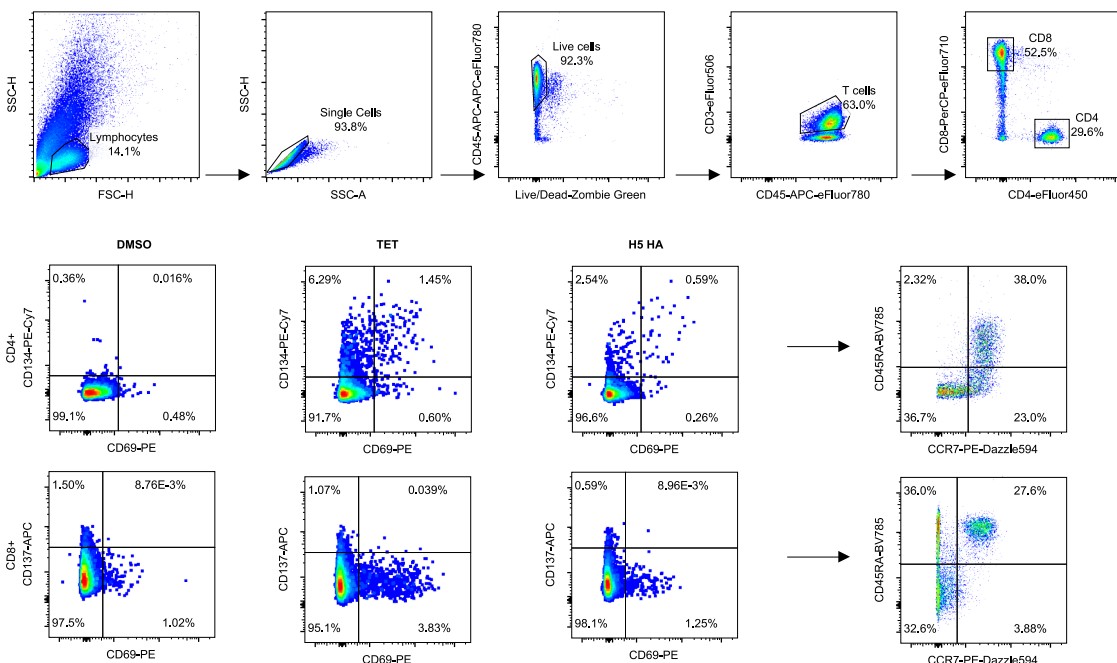

**Extended Data Fig. 2 | Gating strategy for differentiating blood cell populations after flow cytometry.** Representative gating strategy for CD4[+] and CD8[+] T cells after peptide pool stimulation. DMSO, tetanus toxoid (TET) and H5 HA peptide pool stimulated cells are shown as representatives.

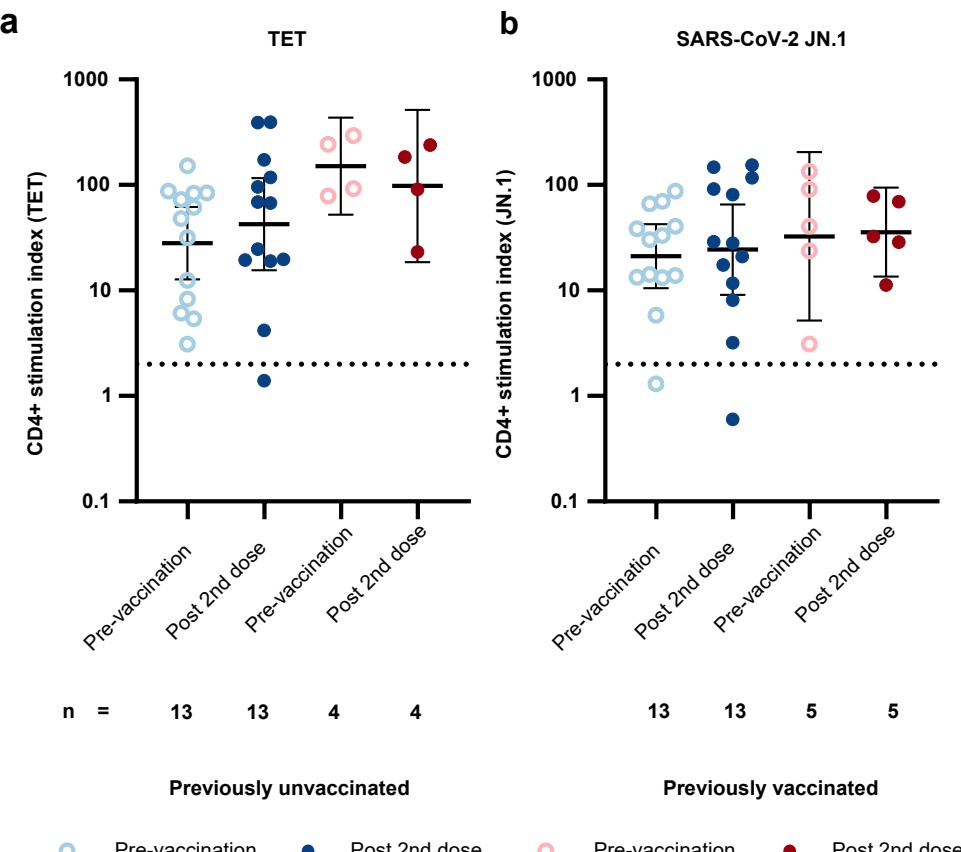

**Extended Data Fig. 3 | CD4⁺ and CD8⁺ T-cell responses specific to TET and SARS-CoV-2 JN.1. a**, Stimulation index fold increases of CD4$^+$/CD69$^+$/CD134$^+$ populations in relation to DMSO. **b**, Stimulation index fold increases of CD8$^+$/ CD69$^+$/CD134$^+$ populations in relation to DMSO. Blue dots indicate individuals with no previous avian influenza vaccinations, and red dots indicate individuals with previous avian influenza vaccinations. The graphs display geometric mean indices (lines) and 95% confidence intervals (whiskers). Dashed line indicates the cut-off threshold. Statistical significance was determined with Wilcoxon matched-pairs signed rank test. Two-sided $p < 0.05$ is considered a significant difference.

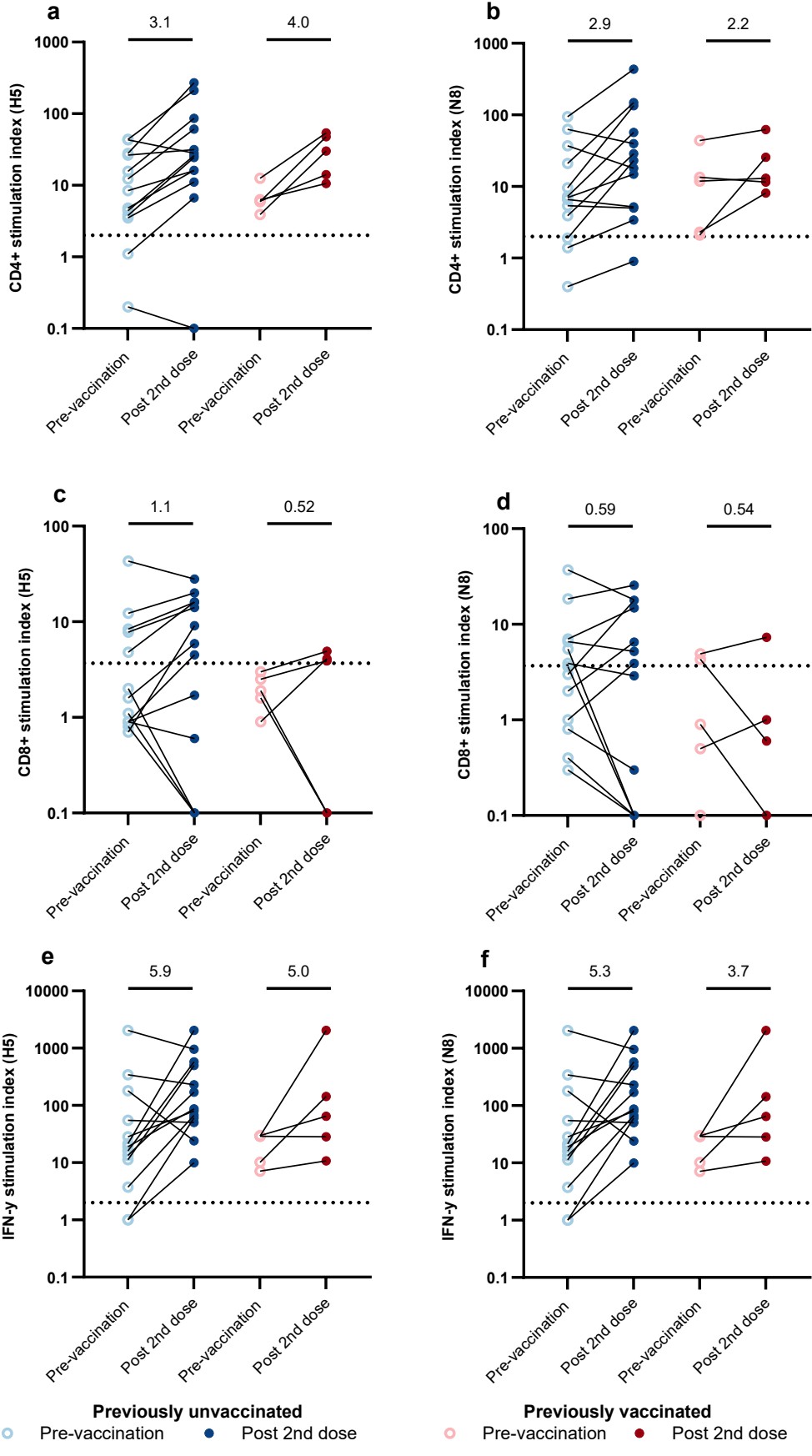

**Extended Data Fig. 4 | See next page for caption.**

**Extended Data Fig. 4 | Kinetics of individual cell-mediated immune responses.**
**a**, Activation induced marked based CD4$^+$/CD69$^+$/CD134$^+$ T-cell responses specific to H5 peptide pool stimulation. **b**, CD4$^+$/CD69$^+$/CD134$^+$ T-cell responses specific to N8 peptide pool stimulation. **c**, CD8$^+$/CD69$^+$/CD134$^+$ T-cell responses specific to H5 peptide stimulation. **d**, CD8$^+$/CD69$^+$/CD134$^+$ cell responses specific to N8 peptide stimulation. Stimulation indices of secreted IFN-γ from PBMC supernatants after stimulation with **e**, H5 and f, N8 peptide pools. Individual responses are shown as lines for two groups, A(H5N1) unvaccinated (n = 13) and previously vaccinated (n = 5), at two time points: pre-vaccination, and three weeks after the second dose. The dashed line indicates the positivity threshold, which was SI 2. The mean fold changes between the samples before vaccination and after the second dose as shown as number on top of the lines.

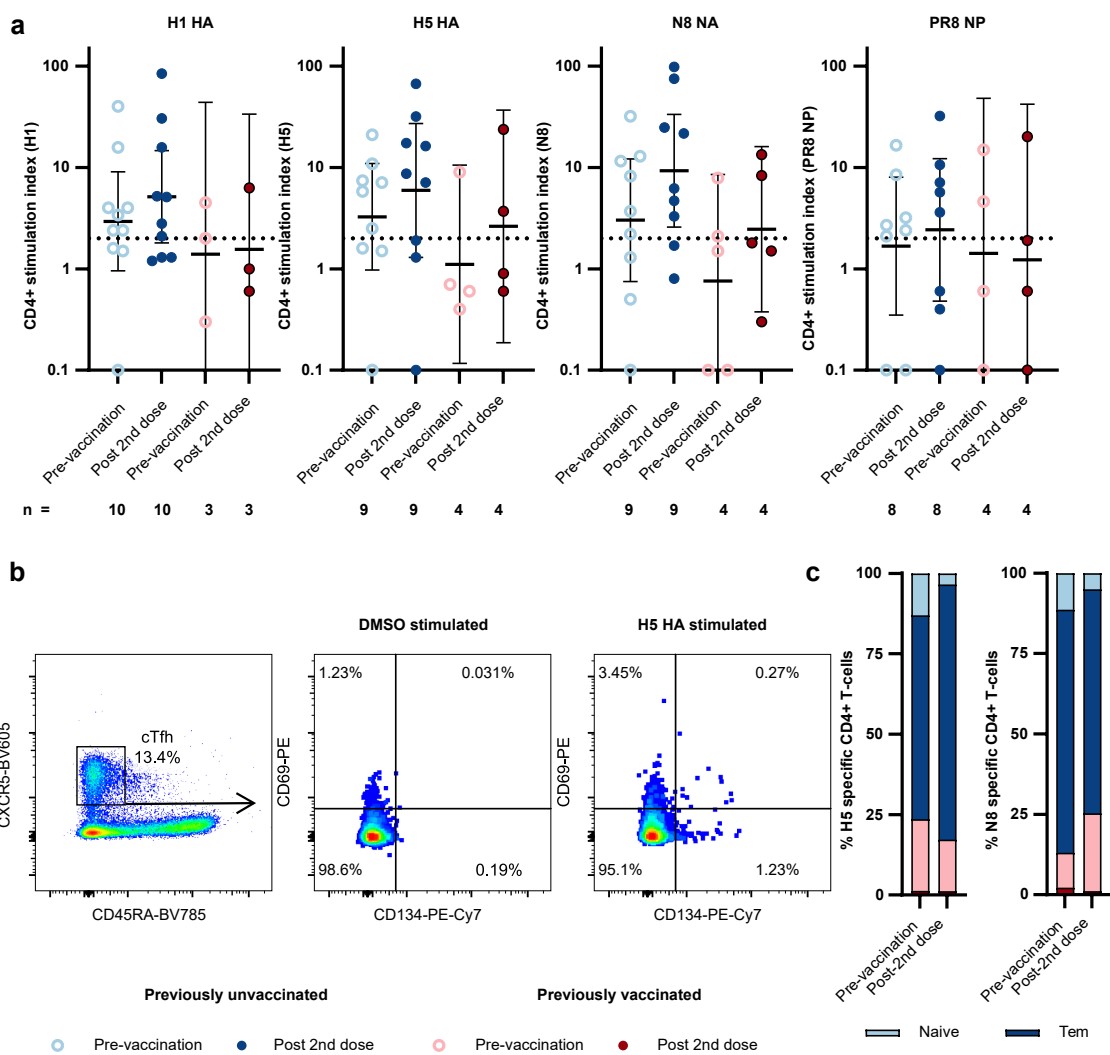

**Extended Data Fig. 5 | Activation induced marker analysis of cTfh responses to influenza peptide stimulation, and distribution of influenza antigen-specific CD4⁺ T cells to naïve, Tem, Tcm, and Temra subclasses. a**, Stimulation indices of H1, H5, N8 and NP peptide pool stimulated CD4⁺/CXCR5⁺/CD45RA⁻/CD69⁺/CD134⁺ T cell populations in relation to DMSO stimulated cells. The graphs display geometric mean indices (lines) and 95% confidence intervals (whiskers). Dashed line indicates the cut-off threshold. Statistical significance was determined with two-sided Wilcoxon matched pairs signed rank test. Two-sided p < 0.05 is considered a significant difference **b**, Gating strategy for identifying CD4⁺/CXCR5⁺/CD45RA⁻/CD69⁺/CD134⁺ follicular Thelper cell (cTfh) populations. **c**, H5-specific CD4+ T cells and N8-specific CD4+ T cells, displayed as percentages of average respective.

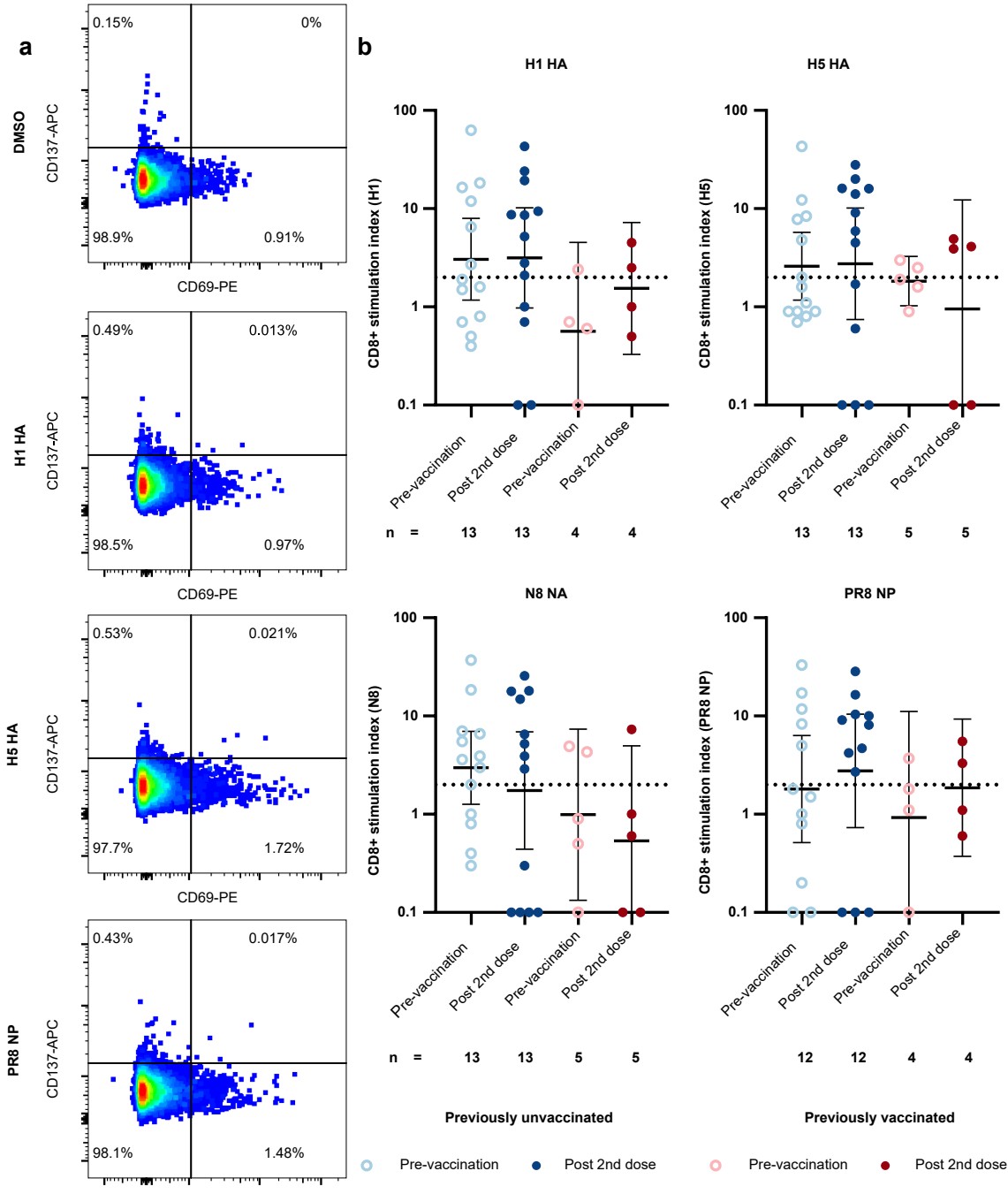

**Extended Data Fig. 6 | Activation induced marker assay of CD8⁺ T-cell responses specific to H1, H5, N8, and NP after stimulating with corresponding peptide pools. a**, Representative gating for identifying CD8⁺/CD69⁺/CD134⁺ population. **b**, Fold increases of antigen-specific CD69⁺ CD134⁺ CD8⁺ T cells in relation to DMSO stimulated cells. In cases when there were no antigen-specific T-cells after DMSO stimulation, DMSO value of the corresponding

Pre-vaccination or Post 2nd dose sample, or value 0.001 was used. Blue dots indicate individuals with no previous avian influenza vaccinations, and red dots indicate individuals with previous avian influenza vaccinations. The graphs display geometric mean indices (lines) and 95% confidence intervals (whiskers). Dashed line indicates the cut-off threshold. Statistical significance was determined with two-sided Wilcoxon matched pairs signed rank test.

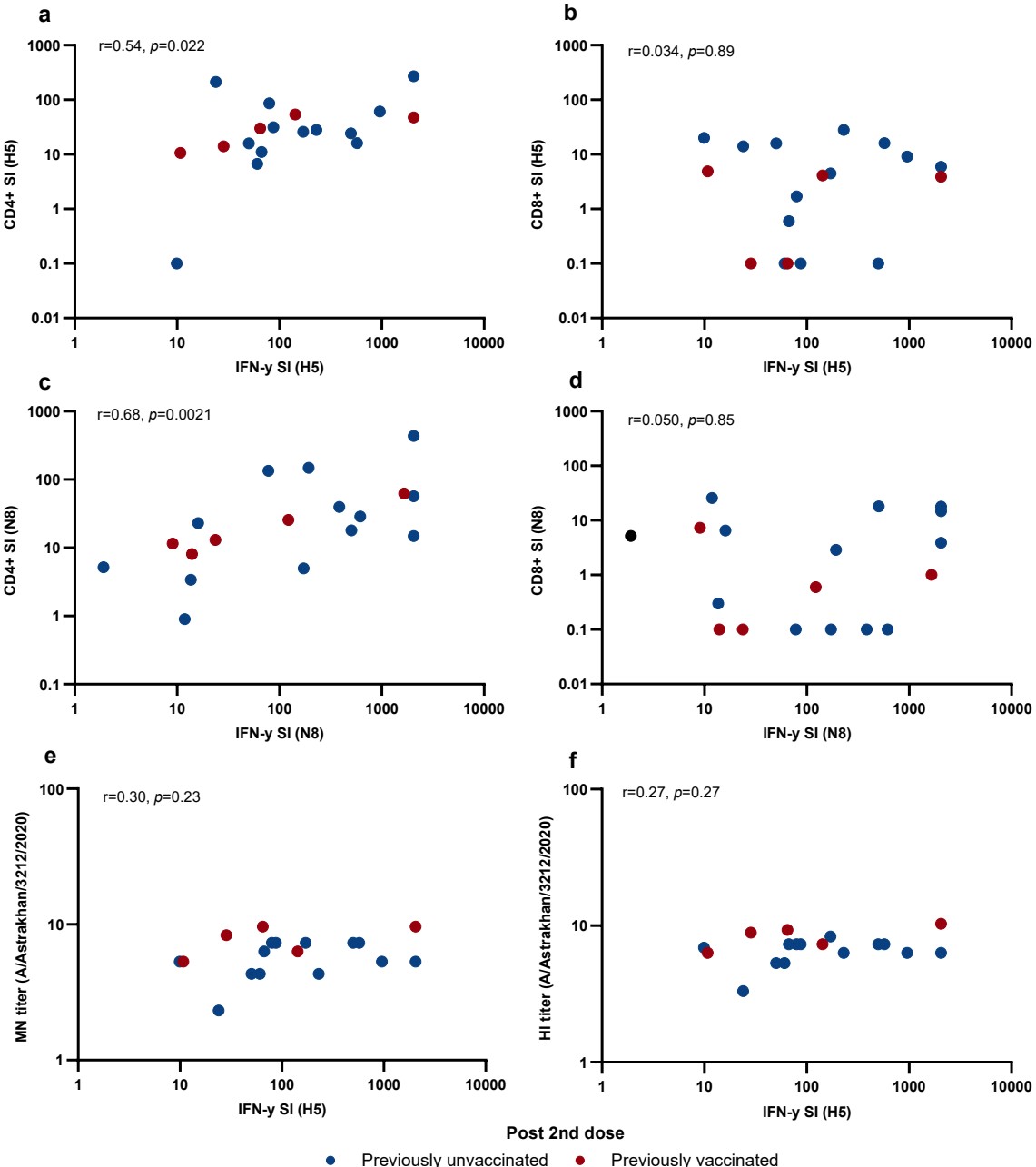

**Post 2nd dose**

● Previously unvaccinated   ● Previously vaccinated

**Extended Data Fig. 7 | Correlation between IFN-γ secretion and CD4⁺ / CD8⁺ T-cell responses and antibody titers.** Spearman's two-sided correlation was used to assess the relationship between the stimulation index (SI) of CD4⁺ and CD8⁺ T cells and IFN-γ SI in response to H5 and N8 peptide pool stimulation. **a**, CD4⁺ SI (H5) correlated positively with IFN-γ SI (H5), **b**, CD8⁺ SI (H5) did not show a significant correlation with IFN-γ SI (H5), **c**, CD4⁺ SI (N8) showed a strong positive correlation with IFN-γ SI (N8), **d**, CD8⁺ SI (N8) showed no correlation with IFN-γ SI (N8). Correlations between IFN-γ SI (H5 peptide pool stimulation) and antibody titers against A/Astrakhan/3212/202 H5N8 virus were also assessed **e**, microneutralization (MN) titer, and **f**, hemagglutination inhibition (HI) titer, neither of which showed significant correlations. Correlation coefficients (r) and corresponding p values (p) are reported in the figure panels. No correction for multiple comparisons was applied.

# Reporting Summary

## Statistics

For all statistical analyses, confirm that the following items are present in the figure legend, table legend, main text, or Methods section.

| n/a | Confirmed | |
|---|---|---|
| ☐ | ☒ | The exact sample size (*n*) for each experimental group/condition, given as a discrete number and unit of measurement |
| ☐ | ☒ | A statement on whether measurements were taken from distinct samples or whether the same sample was measured repeatedly |
| ☐ | ☒ | The statistical test(s) used AND whether they are one- or two-sided<br>*Only common tests should be described solely by name; describe more complex techniques in the Methods section.* |
| ☒ | ☐ | A description of all covariates tested |
| ☐ | ☒ | A description of any assumptions or corrections, such as tests of normality and adjustment for multiple comparisons |
| ☐ | ☒ | A full description of the statistical parameters including central tendency (e.g. means) or other basic estimates (e.g. regression coefficient) AND variation (e.g. standard deviation) or associated estimates of uncertainty (e.g. confidence intervals) |
| ☐ | ☒ | For null hypothesis testing, the test statistic (e.g. *F*, *t*, *r*) with confidence intervals, effect sizes, degrees of freedom and *P* value noted<br>*Give P values as exact values whenever suitable.* |
| ☒ | ☐ | For Bayesian analysis, information on the choice of priors and Markov chain Monte Carlo settings |
| ☒ | ☐ | For hierarchical and complex designs, identification of the appropriate level for tests and full reporting of outcomes |
| ☐ | ☒ | Estimates of effect sizes (e.g. Cohen's *d*, Pearson's *r*), indicating how they were calculated |

*Our web collection on statistics for biologists contains articles on many of the points above.*

## Software and code

Policy information about availability of computer code

| | |
|---|---|
| Data collection | Data was collected in MS Excel version 2408. |
| Data analysis | Data analysis was performed using MS Excel version 2408, GraphPad Prism version 10.2.3 and 10.4.1,R version 4.2.1 and FlowJo version 10.10.0. |

For manuscripts utilizing custom algorithms or software that are central to the research but not yet described in published literature, software must be made available to editors and reviewers. We strongly encourage code deposition in a community repository (e.g. GitHub). See the Nature Portfolio guidelines for submitting code & software for further information.

## Data

Policy information about availability of data

All manuscripts must include a data availability statement. This statement should provide the following information, where applicable:
- Accession codes, unique identifiers, or web links for publicly available datasets
- A description of any restrictions on data availability
- For clinical datasets or third party data, please ensure that the statement adheres to our policy

At the outset of the trial, data-sharing provisions were not included in the informed consent documents signed by participants. In accordance with ethics and institutional policies, we are not authorized to release individual-level or pseudo-anonymized datasets to the public. To protect participant privacy, only de-

identified, aggregated group-level values (without background or individual-level information) are available. These data can be requested from the corresponding author (oona.liedes@thl.fi) and will typically be provided within 2–4 weeks, subject to review for compliance with applicable ethical requirements.

## Research involving human participants, their data, or biological material

Policy information about studies with human participants or human data. See also policy information about sex, gender (identity/presentation), and sexual orientation and race, ethnicity and racism.

| | |
|---|---|
| Reporting on sex and gender | Self-reported gender was not available. Biological sex was inferred from the personal identity code (social security number). However, sex was not used as a variable in the statistical analysis. |
| Reporting on race, ethnicity, or other socially relevant groupings | Race and ethnicity data were not collected. However, participants' occupations were recorded and reported. No analyses were conducted based on occupation or other socially relevant groupings. |
| Population characteristics | The inclusion criteria for the study were (1) age of 18-65 years, (2) belonging to the target group of the avian influenza vaccine, (3) intention to accept at least one dose of the avian influenza vaccine, (4) a native speaker of Finnish, Swedish or English, (5) home address in Finland, (6) ability to give samples three weeks after each dose, (7) preferably the ability to also participate in the follow-up samplings, and (8) a written informed consent. The exclusion criteria were any medical contraindications to influenza vaccination and a history of anaphylactic reaction to any of the constituents or trace residues of the vaccine.<br><br>Participation in this study was voluntary, so some degree of self-selection bias is possible. The study cohort was limited to occupationally exposed individuals, which may not represent the general population. These factors may limit generalizability but are unlikely to affect internal validity of the immunogenicity findings. |
| Recruitment | We invited all registered fur and poultry farmers in the wellbeing services counties of Southern, Central and Northern Ostrobothnia and Kainuu by mail, asking them to forward the invitation to their employees. We approached public sector veterinarians, bird ringers and laboratory workers at the Finnish Food Authority, Finnish Institute for Health and Welfare, Helsinki University Hospital and Diagnostic Center, Turku University Hospital and University of Turku, by sending an information letter of the study, and subsequently an invitation letter to those who expressed their interest to participate in the study. |
| Ethics oversight | The study was conducted in accordance with the Declaration of Helsinki and was authorized by the Finnish Medicines Agency (Fimea) under EU Clinical Trial number 2023-509178-44-00, following evaluation via the EU Clinical Trial Information System (CTIS). |

Note that full information on the approval of the study protocol must also be provided in the manuscript.

# Field-specific reporting

Please select the one below that is the best fit for your research. If you are not sure, read the appropriate sections before making your selection.

☒ Life sciences       ☐ Behavioural & social sciences       ☐ Ecological, evolutionary & environmental sciences

For a reference copy of the document with all sections, see nature.com/documents/nr-reporting-summary-flat.pdf

# Life sciences study design

All studies must disclose on these points even when the disclosure is negative.

| | |
|---|---|
| Sample size | The analysis included 39 adult participants divided into two groups: (1) participants belonging to the target groups for whom the avian influenza vaccine is recommended with no previous influenza (A)H5 vaccination history (n=30) and (2) participants from cohort 1 who have previously received H5 influenza vaccines in 2009, 2011–2012 and/or 2018 (n=9). The targeted sample size of 300 for the study cohort 1 was determined using the sample size formula: $n = (Z^2 \times p(1-p))/E^2$. The calculation was based on a desired 95% confidence level (Z), an assumed seroprotection rate of 75% (p), and a 5% margin of error (E). The result indicates a minimum sample size of 288 subjects required to accurately estimate the proportion of subjects achieving seroprotection. With this sample size, the lower limit of the 95% confidence interval is ≥70%. The number of participants recruited to the study in 2024 remained significantly lower, which will introduce uncertainty into the seroprotection assessment from this sample. |
| Data exclusions | Participants who did not provide blood samples at the scheduled study visits were excluded from the analysis. Additionally, individuals over the age of 65 were excluded, in accordance with the study's inclusion criteria. In the AIM assays, samples with less than 10,000 CD3+ cells were excluded from all analyses, and samples with less than 500 circulating T follicular helper (cTfh) CD4+ cells were excluded from cTfh cell analysis. |
| Replication | MN assay included technical replicates. For FMIA, samples were tested at 1:400 and 1:1600 dilutions in duplicate, with results averaged across four wells. Key experiments were repeated with consistent results, confirming reproducibility. Due to the low number of PBMC samples, there were not enough cells to do replications for the AIM assays. To mitigate this, the AIM assay was optimized beforehand. Positive and negative controls were also included. |
| Randomization | No medical intervention was used in the study and no randomization was applied; all samples meeting the inclusion criteria were analyzed. |

| Blinding | Investigators were blinded to the identity of the participants during all immunological analyses. For FMIA and HI assays, the investigators were also blinded to the timing of the sample collection with respect to vaccination (i.e., whether samples were collected before or after vaccination). For microneutralization and cellular immunity assays, samples from different time points of the same individual were analyzed in parallel within the same run to ensure comparability. Therefore, the timing of the samples (pre- vs. post-vaccination) was known to the investigators conducting these analyses. |
|---|---|

# Reporting for specific materials, systems and methods

We require information from authors about some types of materials, experimental systems and methods used in many studies. Here, indicate whether each material, system or method listed is relevant to your study. If you are not sure if a list item applies to your research, read the appropriate section before selecting a response.

## Materials & experimental systems

| n/a | Involved in the study |
|---|---|
| ☐ | ☒ Antibodies |
| ☐ | ☒ Eukaryotic cell lines |
| ☒ | ☐ Palaeontology and archaeology |
| ☒ | ☐ Animals and other organisms |
| ☐ | ☒ Clinical data |
| ☒ | ☐ Dual use research of concern |
| ☒ | ☐ Plants |

## Methods

| n/a | Involved in the study |
|---|---|
| ☒ | ☐ ChIP-seq |
| ☐ | ☒ Flow cytometry |
| ☒ | ☐ MRI-based neuroimaging |

## Antibodies

| Antibodies used | HRP-conjugated anti-Influenza A antibody (Medix Biochemica, Cat# 100083) and R-PE-conjugated anti-human IgG Fcγ (Jackson ImmunoResearch, Cat# 109-115-098) were used. Fluorochrome-conjugated antibodies used for cytometry were anti-human CD45 (HI30 clone) conjugated with APC-eFluor780 (Invitrogen/Life Technologies, Cat#.47-0459), Anti-human CD3 (UCHT1 clone) conjugated with eFluor506 (Invitrogen/Life Technologies, Cat#. 69-0038-42), Anti-human CD4 (RPA-T4 clone) conjugated with eFluor450 (Invitrogen/Life Technologies, Cat#. 48-0049-42), Anti-human CD8a (SK1) conjugated with PerCP-eFluor710 (Invitrogen/Life Technologies, Cat#. 46-0087-42), Anti-human CD69 (FN50 clone) conjugated with PE (BD Biosciences , Cat#. 555531), Anti-human CD134 (ACT35 clone) conjugated with PE/Cyanine7 (BioLegend, Cat#. 350012), Anti-human CD137 (4B4-1 clone) conjugated with APC (BioLegend, Cat#. 309810), Anti-human CD45RA (HI100 clone) conjugated with Brilliant Violet 785 (BioLegend, Cat#. 304140), Anti-human CD197 (CCR7) (G043H7 clone) conjugated with PE/Dazzle 594 (BioLegend, Cat#. 353236), Anti-human CD185 (CXCR5) (J252D4 clone) conjugated with Brilliant Violet 605 (BioLegend, Cat#. 356930). |
|---|---|
| Validation | The antibodies used in this study were validated by manufacturers. Anti-Influenza A 7307 SPTN-5 (Medix Biochemica, Cat. #100083, monoclonal IgG1) https://www.medixbiochemica.com/anti-influenza-a-100083. Anti-Human IgG, Fcγ (Jackson ImmunoResearch, Cat. #109-115-098, R-PE conjugate, polyclonal) https://www.jacksonimmuno.com/catalog/products/109-115-098. Fluorochrome-conjugated antibodies used for cytometry were anti-human CD45 (HI30 clone) conjugated with APC-eFluor780 (Invitrogen/Life Technologies, Cat#.47-0459) https://www.thermofisher.com/antibody/product/CD45-Antibody-clone-HI30-Monoclonal/47-0459-42, Anti-human CD3 (UCHT1 clone) conjugated with eFluor506 (Invitrogen/Life Technologies, Cat#. 69-0038-42) https://www.thermofisher.com/antibody/product/CD3-Antibody-clone-UCHT1-Monoclonal/69-0038-42, Anti-human CD4 (RPA-T4 clone) conjugated with eFluor450 (Invitrogen/Life Technologies, Cat#. 48-0049-42), https://www.thermofisher.com/antibody/product/CD4-Antibody-clone-RPA-T4-Monoclonal/48-0049-42, Anti-human CD8a (SK1) conjugated with PerCP-eFluor710 (Invitrogen/Life Technologies, Cat#. 46-0087-42), https://www.thermofisher.com/antibody/product/CD8a-Antibody-clone-SK1-Monoclonal/46-0087-42, Anti-human CD69 (FN50 clone) conjugated with PE (BD Biosciences , Cat#. 555531), https://www.bdbiosciences.com/en-fi/products/reagents/flow-cytometry-reagents/research-reagents/single-color-antibodies-ruo/pe-mouse-anti-human-cd69.555531?tab=product_details, Anti-human CD134 (ACT35 clone) conjugated with PE/Cyanine7 (BioLegend, Cat#. 350012), https://www.biolegend.com/en-gb/products/pe-cyanine7-anti-human-cd134-ox40-antibody-7234?GroupID=BLG9043, Anti-human CD137 (4B4-1 clone) conjugated with APC (BioLegend, Cat#. 309810), https://www.biolegend.com/en-gb/products/apc-anti-human-cd137-4-1bb-antibody-3910, Anti-human CD45RA (HI100 clone) conjugated with Brilliant Violet 785 (BioLegend, Cat#. 304140), https://www.biolegend.com/en-gb/products/brilliant-violet-785-anti-human-cd45ra-antibody-7972, Anti-human CD197 (CCR7) (G043H7 clone) conjugated with PE/Dazzle 594 (BioLegend, Cat#. 353236), https://www.biolegend.com/en-gb/products/pe-dazzle-594-anti-human-cd197-ccr7-antibody-9811, Anti-human CD185 (CXCR5) (J252D4 clone) conjugated with Brilliant Violet 605 (BioLegend, Cat#. 356930), https://www.biolegend.com/en-gb/products/brilliant-violet-605-anti-human-cd185-cxcr5-antibody-12362. |

## Eukaryotic cell lines

Policy information about cell lines and Sex and Gender in Research

| Cell line source(s) | MDCK cell lines (ATCC-CCL-34 and ATCC-CRL-2935) were obtained from the American Type Culture Collection (ATCC). |
|---|---|
| Authentication | Authentication was performed by the supplier. |
| Mycoplasma contamination | Tested and negative for Mycoplasma. |

| Commonly misidentified lines (See ICLAC register) | No misidentified lines were used. |
|---|---|

# Clinical data

Policy information about clinical studies

All manuscripts should comply with the ICMJE guidelines for publication of clinical research and a completed CONSORT checklist must be included with all submissions.

| Clinical trial registration | The study was registered in the EU Clinical Trials Information System (CTIS) under the EU CT number 2023-509178-44-00 on 19 April 2024. |
|---|---|
| Study protocol | The study was conducted in accordance with the standards of Good Clinical Practice, the Declaration of Helsinki and local legal and regulatory requirements. The full study protocol is available in the EU Clinical Trials Information System (CTIS) as part of the trial registration (EU CT number: 2023-509178-44-00). The study has received authorization from the Finnish Medicines Agency Fimea. Written informed consent to participate was obtained from all participants before sampling. Participation in the study was voluntary and uncompensated. |
| Data collection | Participants were recruited in Finland starting in June 2024. Recruitment targeted individuals for whom the MF59-adjuvanted A(H5N8) influenza vaccine (clade 2.3.4.4b A/Astrakhan/3212/2020, Seqirus) was recommended—specifically those at risk of exposure to infected animals, including fur and poultry farm workers, veterinarians, bird ringers, and laboratory personnel handling avian influenza viruses or potentially contaminated samples. Sample collection was conducted at laboratory centers within the well-being services counties. Participants were asked to indicate their professional affiliation or vaccine target group. |
| Outcomes | Primary outcome measure was seroconversion proportion after the second vaccine dose. Secondary outcomes were pre-defined in the study protocol and registered in EU Clinical Trial Registry (2023-509178-44-00). |

# Plants

| Seed stocks | *Report on the source of all seed stocks or other plant material used. If applicable, state the seed stock centre and catalogue number. If plant specimens were collected from the field, describe the collection location, date and sampling procedures.* |
|---|---|
| Novel plant genotypes | *Describe the methods by which all novel plant genotypes were produced. This includes those generated by transgenic approaches, gene editing, chemical/radiation-based mutagenesis and hybridization. For transgenic lines, describe the transformation method, the number of independent lines analyzed and the generation upon which experiments were performed. For gene-edited lines, describe the editor used, the endogenous sequence targeted for editing, the targeting guide RNA sequence (if applicable) and how the editor was applied.* |
| Authentication | *Describe any authentication procedures for each seed stock used or novel genotype generated. Describe any experiments used to assess the effect of a mutation and, where applicable, how potential secondary effects (e.g. second site T-DNA insertions, mosiacism, off-target gene editing) were examined.* |

# Flow Cytometry

## Plots

Confirm that:

☒ The axis labels state the marker and fluorochrome used (e.g. CD4-FITC).

☒ The axis scales are clearly visible. Include numbers along axes only for bottom left plot of group (a 'group' is an analysis of identical markers).

☒ All plots are contour plots with outliers or pseudocolor plots.

☒ A numerical value for number of cells or percentage (with statistics) is provided.

## Methodology

| Sample preparation | Sample preparation is detailed in the 'Activation Induced Marker (AIM) Assay and Flow Cytometry' subsection of the 'Materials and Methods' section. |
|---|---|
| Instrument | LSRFortessa (BD Bioscience) |
| Software | FlowJo 10.10.0 |
| Cell population abundance | Cells were seeded at 1,000,000 cells per well, with a minimum of 10,000 $CD3^+$ T cells and over 500 circulating T follicular helper (cTfh) cells required for downstream analysis. |
| Gating strategy | Gating strategy is shown in the Extended Data Figure 2. |

☒ Tick this box to confirm that a figure exemplifying the gating strategy is provided in the Supplementary Information.

