## [Peer Review File · Nature Microbiology]

Influenza A(H5N8) Vaccine Induces Humoral and Cell-mediated Immunity Against Highly Pathogenic Avian Influenza Clade 2.3.4.4b A(H5N1) Viruses

Corresponding Author: Ms Oona Lieder

Version 0:

Reviewer comments:

Reviewer #1

(Remarks to the Author)

The authors addressed all of the comments raised by the initial review.

Reviewer #2

(Remarks to the Author)

The authors addressed my comments and questions.

Decision Letter:

Our ref: NMICROBIOL-25082918-T

12th September 2025

Dear Oona,

Thank you for submitting your revised manuscript "Influenza A(H5N8) Vaccine Induces Humoral and Cell-mediated Immunity Against Highly Pathogenic Avian Influenza Clade 2.3.4.4b A(H5N1) Viruses" (NMICROBIOL-25082918-T). It has now been seen by the original referees and their comments are below. The reviewers find that the paper has improved in revision, and therefore we'll be happy in principle to publish it in Nature Microbiology, pending minor revisions to satisfy the referees' final requests and to comply with our editorial and formatting guidelines.

Thank you again for your interest in Nature Microbiology Please do not hesitate to contact me if you have any questions.

Sincerely,

Reviewer #1 (Remarks to the Author):

The authors addressed all of the comments raised by the initial review.

Reviewer #2 (Remarks to the Author):

The authors addressed my comments and questions.

Version 1:

Decision Letter:

9th October 2025

Dear Oona,

I am pleased to accept your Article "Influenza A(H5N8) Vaccine Induces Humoral and Cell-mediated Immunity Against Highly Pathogenic Avian Influenza Clade 2.3.4.4b A(H5N1) Viruses" for publication in Nature Microbiology. Thank you for having chosen to submit your work to us and many congratulations.

Authors may need to take specific actions to achieve compliance with funder and institutional open access mandates. If your research is supported by a funder that requires immediate open access (e.g. according to <https://www.springernature.com/gp/open-science/plan-s-compliance> or the [Plan S principles](https://www.springernature.com/gp/open-science/us-federal-agency-compliance) or the [NIH public access policy](https://www.springernature.com/gp/open-science/us-federal-agency-compliance)) then you should select the gold OA route, and we will direct you to the compliant route where possible. Because authors warrant under our subscription licensing terms that they haven't committed to licensing any version of their article under a licence inconsistent with the terms of our agreement – including the applicable embargo period – publication under the subscription model isn't suitable for authors whose funders require no embargo.

To assist our authors in disseminating their research to the broader community, our SharedIt initiative provides you with a unique shareable link that will allow anyone (with or without a subscription) to read the published article. Recipients of the link with a

subscription will also be able to download and print the PDF.

Congrats again to you and your co-authors! I am looking forward to seeing your paper published.

With kind regards,

P.S. Click on the following link if you would like to recommend Nature Microbiology to your librarian
<http://www.nature.com/subscriptions/recommend.html#forms>

** Visit the Springer Nature Editorial and Publishing website at http://editorial-jobs.springernature.com?utm_source=ejP_NMicro_email&utm_medium=ejP_NMicro_email&utm_campaign=ejp_NMicro for more information about our career opportunities. If you have any questions please click [here](mailto:editorial.publishing.jobs@springernature.com).

Reviewers' Comments:

Reviewer #1 (Remarks to the Author):

This paper supplies the first human data on the MF59-adjuvanted H5N8 vaccine, captured in real time during Finland's 2024 H5N1 outbreak, and demonstrates that two doses (or a single booster in previously primed adults) induce broad neutralising and CD4⁺ responses against emergent clade 2.3.4.4b strains, offering rare, actionable evidence for pandemic stockpile use and dose-sparing policies. Laboratory assays are sound and software versions are documented, but sample-size shortfall, representativeness concerns, missing-data handling and lack of multiplicity control currently limit the quantitative strength of the conclusions. Implementing the concrete revisions above would bring the manuscript in line with statistical-rigour expectations.

Study design & cohorts

Comment: 1. The study planned to enrol 288 H5-naïve adults so that the seroprotection rate (SPR) could be estimated within ± 5 percentage points. In reality, only 30 naïve adults (antibody endpoints) and 13 naïve adults (T-cell endpoints) were analysed, barely one-tenth of the target. The manuscript should instead highlight and interpret the much wider CIs in both text and figures. T-cell endpoints are severely under-powered ($n = 18$ total). What is the reason for this?

Response: We thank the reviewer for this important observation. We acknowledge that the reduced sample size compared to the original target has resulted in relatively wider confidence intervals (CIs) for both antibody and T-cell endpoints. The protocol included a sample size calculation for the primary objective, aiming to estimate the SPR with a $\pm 5\%$ margin of error at a 95% confidence level, as the reviewer points out. The study hypothesis was that a two-dose vaccination schedule would induce seroprotection in at least 70% of vaccinees, meaning that the lower bound of the two-sided 95% confidence interval for seroconversion would meet or exceed 70%. Based on an assumed SPR of 75%, the target sample size was 288 participants, ensuring this confidence criterion. Although the planned enrolment was not met and only 30 H5-naïve adults were analyzed, the observed SPR was considerably higher than anticipated. Consequently, the lower bound of the 95% confidence interval still met or exceeded 70% (95% CI 70–97% by microneutralization assay and 90–100% by hemagglutination inhibition assay against the vaccine virus), thereby fulfilling the predefined protocol criteria for the primary objective.

The final number of participants analyzed was lower than planned due to recruitment challenges. Recruitment was most notably hindered by the very low uptake of the vaccine among the target groups to whom it was recommended. Despite our best efforts and multiple outreach attempts, particularly towards fur farm workers, we were unable to recruit representatives from all target groups due to difficulties in reaching individuals and the notably low interest in vaccine uptake within this group. The limited sample size for the T-cell analyses ($n=18$) is primarily due to two factors. First, not all participants consented to donate blood for PBMC isolation, as this requires a significantly larger blood volume than serum sampling. Second, PBMC samples could only be collected from participants in the Helsinki metropolitan and Turku areas, as same-day isolation was

required. Additionally, the total number of PBMCs isolated per participant varied, and in some cases, the yield was insufficient to carry out the planned analyses.

To clarify the number of samples included and reasons for exclusion, we have refined the description of the PBMC samples in lines 90 (“28 participants”) and 93 (“In addition, the amount of isolated PBMCs from 7 participants was insufficient for analysis”).

Representativeness & potential bias

Comment: 2. Participants were 60 % laboratory staff and 0 % fur-farm workers, i.e. not the principal risk group the vaccine targets. The discussion should state that immune responses may differ in the intended population (older, more male, different exposure history).

Response: We acknowledge that laboratory workers constituted the largest single occupational group among the participants in our study, while no fur-farm workers participated. As the study was voluntary, participation could not be mandated. We made efforts to reach out to all Finnish fur farmers in the regions where the study was conducted, but none chose to participate. It is important to note that the national vaccination recommendation in Finland does not define a single primary risk group. Instead, it includes a broader range of individuals aged 18 or over who, due to their work or other circumstances, are at increased risk of contracting avian influenza. This includes not only fur animal and poultry workers but also official bird ringers, veterinarians and laboratory personnel handling potentially infectious samples. We agree that immune responses may vary across different subgroups within the target population, particularly considering factors such as age and sex, and potentially, exposure history. We have studied individuals exposed in fur farms where avian influenza H5N1 was detected in 2023, in a separate study, currently unpublished study. In that study, we did not detect any seropositive cases for A/Astrakhan/3212/2020 among the exposed individuals, suggesting that exposure history does not necessarily influence antibody responses. Due to the limited sample size in the present study, we are unable to draw reliable conclusions regarding the potential effects of age and gender on antibody responses. We have excluded individuals over 65 years of age from the study population to reduce confounding by age-related changes in immune function.

Comment: 3. Analyses were restricted to those who provided all samples in the correct sequence. This complete-case approach can bias estimates if missingness relates to response; a sensitivity analysis using all available data (e.g. mixed-effects models with maximum-likelihood estimation) is recommended.

Response: We thank the reviewer for this valuable suggestion. The pre-defined primary outcome measure described in the study protocol was the SPR, assessed three weeks after the second dose, administered at least 21 days after the first dose, in subjects aged 18 to 65 years who belong to the target groups for avian influenza vaccination. In this study, we excluded from the analyses any samples that were not collected according to the study protocol. This included participants over 65 years of age and those whose samples were not taken at the correct time points in relation to the vaccinations (for example cases where the first sample was collected after the initial vaccination, rather than prior to vaccination). Figure 1 details the number of excluded participants and the corresponding reasons. These exclusions were made to ensure consistency and protocol adherence across the dataset. We appreciate your suggestion regarding the inclusion of all

suitable samples and using mixed-effects models with maximum-likelihood estimation, that to our knowledge do not require complete cases, and will take this into account in future publications with longer-term follow-up samples (for example, in cases where the participant has only provided a 6-month sample).

Statistical tests & modelling

4. Distribution checks were performed with Shapiro–Wilk; non-normal data were \log_2 -transformed and analysed with non-parametric tests (Mann–Whitney, Wilcoxon). This is acceptable for small samples, but repeated-measures designs are better handled with paired linear mixed-effects models, which use all time-points simultaneously and provide effect sizes with CIs.

Response: We appreciate the suggestion to consider paired linear mixed-effects models (LMMs) for repeated-measures analysis. While LMMs offer flexibility in modeling inter-individual variability and time interactions, our analysis approach was guided by the specific research questions and the structure of the data. Rather than conducting repeated-measures comparisons across all timepoints, we focused on predefined pairwise comparisons. Accordingly, we used the Wilcoxon signed-rank test for paired samples and the Mann–Whitney U test for unpaired comparisons, both of which are non-parametric methods suitable for our non-normally distributed data. These methods allowed us to address the key hypotheses without assuming normality.

5. Multiple comparisons: dozens of endpoints (MN, HI and FMIA titres against three viruses at three time points; several T-cell read-outs) were tested without adjustment. Even in an exploratory study FDR or Bonferroni-Holm control should be reported, or results should be clearly labelled as descriptive.

Response: We appreciate the reviewer's observation regarding multiple comparisons. We have added a clarifying statement to the manuscript indicating that the results are descriptive in line 558 ("All results are presented as descriptive; statistical tests were performed without adjustment for multiple comparisons").

6. Correlation analyses between MN and HI titres used Spearman ρ and simple linear regression ($R^2 = 0.76$). With only 142 observations, many not independent because each participant contributes three sera, standard errors are underestimated. Bootstrapping clustered by participant or using GEE would yield more reliable CIs.

Response: We thank the reviewer for this insightful comment. We agree that bootstrapping clustered by participant or using generalized estimating equations (GEE) would yield a more robust estimate if the goal were to obtain a highly precise point estimate. However, given the aim of this specific analysis, to determine the MN titer that corresponds to an HI titer of 40, we consider our current method sufficient. While the dataset includes repeated measurements from the same individuals at different time points, this does not affect the interpretation in this context.

Interpretation

7. The manuscript states that “these results demonstrate that the vaccine likely provides cross-protection” (Abstract line 31). Given the small, self-selected cohort and absence of efficacy data, this phrasing should be tempered (e.g. “suggests immunogenicity consistent with potential cross-protection”).

Response: We thank the reviewer for this suggestion. We have updated text in line 31 (“These results indicate immunogenicity consistent with potential cross-protection against circulating H5 clade 2.3.4.4b viruses”).

8. The authors correctly mention the imprecision caused by small numbers, but the Discussion could further explore how prior avian influenza vaccination, sex and age (27–63 y in naïve vs 40–61 y in primed) might confound comparisons.

Response: We thank the reviewer for this insightful comment. As the SPR was calculated separately for the naïve and primed groups, prior avian influenza vaccination does not confound these comparisons. The impact of prior vaccination was already discussed in the manuscript in lines 241-252. (“In another study, individuals primed six years earlier with an MF59-adjuvanted A/duck/Singapore/1997 (clade 0-like) developed higher frequencies of memory B cells and rapidly produced high titers of neutralizing antibodies against diverse A(H5N1) clades after receiving a A/Vietnam/1194/2004 (clade 1) vaccine. These findings suggest that distant priming can establish a pool of memory B cells that respond robustly to mismatched vaccines years later. Similarly, in our earlier work, two primary doses generated strain-specific responses, while a later heterologous dose boosted cross-clade antibody levels. We found that in participants who had previously been vaccinated with different H5 vaccines, a single dose of the current vaccine elicited a strong antibody response, with no additional boost from a second dose. This aligns with our earlier findings showing that a second dose given shortly after the first does not significantly enhance antibody levels, whereas a dose given after a longer interval does”). Regarding age and sex, the age ranges between the groups (27–63 years in naïve vs. 40–61 years in primed) are not substantially different relative to the sample size, and we do not expect them to meaningfully influence the results.

Reviewer #2 (Remarks to the Author):

Liedes et al evaluate humoral and cellular immune responses elicited by an adjuvanted clade 2.3.4.4b H5 vaccine in humans. They find that the vaccine elicits neutralizing antibodies in the majority of participants in the study, with higher antibody levels in individuals who were previously vaccinated with H5. The authors also examined T cell responses, although those data were not as compelling since most participants possessed H5, N8, and NP reactive T cells prior to vaccination and there were only limited numbers of donors who contributed PBMCs for cellular analyses. Overall, while the data are not surprising (i.e., seroconversion following vaccination with novel H5 antigen), the study is important because it suggests that a clade 2.3.4.4b H5 vaccine could be used to minimize risk of H5 infection in humans.

1. It would be interesting to complete studies to determine if the specificity of H5-reactive antibodies is different in the group that was previously H5 vaccinated versus the group that was not. For example, previous studies showed that individuals exposed to H5/2004 for the 1st time primarily produce antibodies that target the HA stalk (conserved in H2 and H1 viruses). After a 2nd H5/2004 immunization, those individuals then started to produce antibodies targeting the HA head (see studies by Patrick Wilson et al.). The authors should consider experiments to determine if different levels of HA head versus stalk antibodies are elicited after a single vaccination versus a 2nd vaccination in individuals with and without prior H5 vaccine history.

Response: We thank the reviewer for the thoughtful suggestion regarding the analysis of HA head versus stalk antibody responses in participants with and without prior H5 vaccination. In this study, we employed key methods outlined in our study protocol to address our primary research question: whether vaccination induces an antibody response in participants without prior H5 vaccination. A secondary objective, assessing humoral and cellular immune responses in individuals with prior avian influenza vaccination, was incorporated after it became apparent during recruitment that some participants had a history of H5 vaccination. While the reviewer's suggested experiments are scientifically valuable, they extend beyond the scope of the current study. We intend to investigate cross-protection and antibody specificity in future work, at which point approaches such as those proposed may be considered.

2. A recent *Nature Medicine* paper demonstrated that individuals with different birth year and seasonal influenza exposure histories have different levels of pre-existing H5 reactive antibodies prior to vaccination. It would be interesting to graph titers and responses in the current study as a function of birth year.

Response: We thank the reviewer for highlighting the recent findings on the influence of birth year and seasonal influenza exposure history on pre-existing H5-reactive antibodies. In our study, no pre-existing H5 antibody responses to the A/Astrakhan/3212/2020 virus were detectable by microneutralization assay in participants who had not previously received H5 vaccines. As suggested, we plotted both pre-vaccination and post-vaccination antibody titers measured with microneutralization test as a function of birth year; however, no clear dependence or trend was observed. We note that the age distribution of our study population was relatively narrow (birth years ranging from approximately 1961 to 1997), which may limit the ability to detect such associations and makes the data suboptimal for evaluating birth year-related trends.

3. It seems surprising that 6 of 30 individuals had detectable H5 HI antibodies prior to vaccination.

Response: We acknowledge that the presence of detectable H5 HI antibodies in 6 of 30 participants prior to vaccination may seem surprising. However, we have addressed this in the Discussion section in line 226 ("*We observed a strong correlation between the MN and HI titers ($r=0.89$) indicating that both assays measure functional neutralizing antibodies despite differing sensitivities in SPR assessment*"). We also discuss how methodological factors, including the use of horse erythrocytes in the HI assay, can influence sensitivity and result in higher HI titers

compared to MN, which contrasts with earlier findings suggesting MN is typically more sensitive in lines 230-235 (*“Neutralizing A(H5N1) antibody responses have usually confirmed the trend seen in the HI assay, however, the results do not always correlate. Increased sensitivity and higher HI antibody responses have been observed using horse erythrocyte-based HI assay when compared to using turkey or chicken erythrocytes when performing serology of human sera against avian influenza viruses. Accordingly, we found that HI titers against the vaccine antigen were higher than those measured with the MN assay, which contradicts earlier findings that have suggested that the MN assay is a more sensitive method than the HI assay”*). These considerations help explain the observed pre-vaccination HI titers and support the robustness of our serological assessments.

4. *It is difficult to draw conclusions on the T cell data since there are a limited number of participants and a lot of pre-existing responses.*

Response: We acknowledge the reviewer’s concern regarding the limited number of participants and the presence of pre-existing T-cell responses. As noted in the Discussion lines 288-295 (*“The presence of pre-vaccination H5-specific T-cell responses indicate that cross-reactive T cells pre-exist in the general population. A recent study on T cell epitope analysis of A(H5N1) clade 2.3.4.4b suggested that conserved epitopes may enable pre-existing immunity to attenuate the severity of A(H5N1) infections in humans. The authors demonstrated that prior seasonal influenza infections have seeded a broad pool of cross-reactive memory T cells, with approximately 70% of catalogued CD4⁺ and 60% of CD8⁺ epitopes being ≥90% conserved in circulating clade 2.3.4.4b viruses. Notably, CD4⁺ responses were more pronounced than CD8⁺ responses in a combined AIM and ICS assay, in line with our findings”*), pre-existing T-cell immunity has been observed in this and other studies. However, despite these baseline responses, we were able to demonstrate clear vaccine-induced increases in T-cell activity.

Minor:

line 50: there have now been multiple documented H5 exposures in US dairy cattle

Response: We agree with the reviewer that multiple H5 spillover events have been documented in U.S. dairy cattle since spring 2024. However, our manuscript does not aim to cover the full extent of these events. Rather, it was essential for this publication to describe where and how the outbreak began, in order to provide context for the importance of studying vaccine-induced immune responses in the face of an emerging zoonotic threat

Reviewer #3 (Remarks to the Author):

In the manuscript entitled “Inactivated Zoonotic Influenza A(H5N8) Vaccine Induces Humoral and Cell-mediated Immune Responses 2 Against Highly Pathogenic Avian Influenza Clade 2.3.4.4b A(H5N1) Viruses” the authors characterize the immune response in animal handlers and farm workers who received the H5N8 vaccine due to increased risk of potential exposure to the H5N1 virus in Finland. As workers with increased risk of exposure to H5N1 in Finland were vaccinated

with the inactivated MF59 adjuvanted H5N9 vaccine, they were invited to participate in the study where they had blood and PBMC samples collected for immunological analysis. 52 participants enrolled in the study, in the end, 30 participants who had not previously received an H5-containing vaccine, and 9 participants who had previously received an H5-containing vaccine were included in the analysis. Although a smaller number than they had hoped participated, the data was such that there were significant difference in the groups.

The team found that the H5N8 vaccine induced antibodies that targeting multiple 2.3.4.4.b clade H5N1 viruses, and that individuals who had been previously vaccinated with H5N1 vaccines (2-6 previous doses), had immunological memory and so responded with a significantly higher antibody response than those who had not been vaccinated before. In addition to the antibody responses, they demonstrated strong CD4 but not CD8 responses to the inactivated vaccines.

This data is novel and provides robust evidence that the H5N8 vaccine should provide adequate protection against similar clade viruses. The approach is adequate, assessing both antibody and cell-mediated immune responses to a number of circulating influenza viruses.

Response: We thank the reviewer for this encouraging comment. We are pleased that the novelty of the data and the comprehensive assessment of both humoral and cell-mediated immune responses were appreciated. We believe these findings contribute valuable insights into the potential cross-protective efficacy of the H5N8 vaccine.